# Phytoplankton community succession and biogeochemistry in a bloom simulation experiment at an estuary-ocean interface

Jenna A. Lee[1], Joseph H. Vineis[1,§], Mathieu A. Poupon[1,2], Laure Resplandy[1,3], Bess B. Ward[1,3]

[1]Department of Geosciences, Princeton University, Princeton, NJ, 08544, USA

[2]Atmospheric and Oceanic Sciences Program, Princeton University, Princeton, NJ, 08544, USA

[3]High Meadows Environmental Institute, Princeton University, Princeton, NJ, 08544, USA

[§]*Current affiliation*: Marine Biological Laboratory, University of Chicago, Woods Hole, MA, 02543, USA

*Correspondence to*: Jenna A. Lee (jennaal@princeton.edu)

**Abstract.** Phytoplankton blooms, especially diatom blooms, account for a large fraction of marine carbon fixation. Species
succession and biogeochemical parameters change rapidly over a bloom, and determine the resulting biological productivity. This study implemented daily sampling of a 24–L microcosm bloom simulation experiment to assess changes in assemblage and biogeochemical processes while excluding changes due to advection. $^{15}NO_3^-$ and $H^{13}CO_3^-$ tracer incubations were performed alongside pigment and DNA sampling to compare temporal trends in community composition and primary productivity (nitrogen (N) and carbon (C) transport rates). Rapid drawdown of nutrients and maximum C and N transport rates
corresponded with peak chlorophyll–a and fucoxanthin pigment concentrations. Fucoxanthin, typically associated with diatoms, was the dominant diagnostic pigment, with very low peridinin (dinoflagellate) and zeaxanthin (cyanobacteria) concentrations, indicating a diatom bloom. 18S rRNA gene analysis showed clear community succession throughout the duration of the bloom and multiple species of diatoms co–occurred, including during the bloom peak. The presence of metazoan 18S, high carbon–to–chlorophyll ratios, and a model analysis provide evidence of grazing in the latter half of the
bloom. A traditional bloom framework suggests that species succession occurs as the bloom progresses and that phytoplankton diversity reaches a minimum of just one or two dominant species when phytoplankton productivity is at its maximum. However, this study produced a negatively monotonic productivity–diversity relationship with relatively high minimum diversity values. This 18S–based analysis therefore presents a more complex relationship between bloom progression and phytoplankton diversity.

## Short summary

Concurrent sampling of environmental parameters, productivity rates, photopigments, and DNA were used to analyze 24–L estuarine diatom bloom microcosms. Biogeochemical data and an ecological model indicated that the bloom was terminated by grazing. Comparisons to previous studies revealed (1) additional community and diversity complexity using 18S amplicon
vs. traditional pigment–based analyses, and (2) a potential global productivity–diversity relationship using 18S and carbon transport rates.

# 1 Introduction

Marine primary productivity is dominated by phytoplankton and accounts for nearly half of global net carbon fixation (Field et al., 1998). Phytoplankton blooms are of particular importance because their communities are often dominated by large phytoplankton (e.g., diatoms) which can grow rapidly. Diatoms are unique phytoplankton not only because of their large size and high maximum growth rates, but also because they have silica cell walls which may reduce grazing pressure (Grønning and Kiørboe, 2020) and increase their sinking rate (Legendre and Le Fèvre, 1995), leading them to contribute significantly to marine carbon export from the surface ocean (Jin et al., 2006). Diatoms that do not escape grazing become important food sources which support productive fisheries (Jackson et al., 2011) through highly efficient trophic energy transfer, either via large zooplankton (Sommer et al., 2002) or directly to fish (Ryther, 1969; Van Der Lingen, 2002; Costa and Garrido, 2004). Much of the research on diatom blooms is focused on upwelling systems, but the shallow, high nutrient environment of estuaries also supports annual diatom blooms and large fisheries (Marshall et al., 2005; Harding et al., 2015; Bilkovic et al., 2019). Estuarine systems serve as a buffer between land and the open ocean, modulating the type and amount of carbon that is exported to continental shelves, regions which are responsible for high carbon burial and export to the deep ocean (Bauer et al., 2013). Thus, understanding phytoplankton communities and productivity in estuarine systems is key to understanding global nutrient cycling.

The productivity–diversity relationship (PDR) is used to investigate complex mechanisms such as species interactions, coexistence, and overall drivers of community composition. However, the PDR of marine microbial communities varies greatly between studies and is heavily dependent on study region and methods used for determining diversity and productivity (Smith, 2007; Graham and Duda, 2011). The canonical view of bloom diversity and succession is that initial high nutrient conditions allow fast–growing phytoplankton to outcompete other species, leading to a minimum community diversity when phytoplankton biomass or rates of productivity are at their peak (Irigoien et al., 2004). Blooms are then terminated by a combination of bottom–up factors like nutrient depletion, and top–down factors such as grazing and viral and parasitic infection, transitioning into a more diverse microbial community.

Chesapeake Bay is the largest estuary in the United States (Bilkovic et al., 2019) and is often used as a model system for a broader understanding of coastal environments. Many studies have documented the seasonal succession of the phytoplankton community in the bay, primarily using pigment– and microscopy–based analyses (Marshall et al., 2005; Adolf et al., 2006; Harding et al., 2015). Such studies generally agree on the year–round importance of diatoms; diatoms dominate the phytoplankton community in the spring and fall while a more diverse phytoplankton community composed of diatoms, dinoflagellates, and cyanobacteria is present in the summer. Though there have been 18S–based studies in the tributaries of Chesapeake Bay, especially of harmful dinoflagellate bloom communities, there are very few reported metabarcoding studies of the eukaryotic community in the main stem of Chesapeake Bay (Cram et al., 2024; Wang et al., 2024). Molecular techniques

like metabarcoding can detect a much greater diversity of marine eukaryotes than microscopy or pigment analyses, including species that are not possible to detect or distinguish visually (López–García et al., 2001; Massana and Pedrós–Alió, 2008; Xu et al., 2023).

Here we compare the composition and PDR of an estuarine diatom bloom with both the historical trends in Chesapeake Bay and with patterns observed globally in the open ocean. We utilized daily sampling of 18S amplicons, pigment concentrations, carbon (C) and nitrogen (N) transport rates, and particulate organic matter (POM) accumulation in carboy incubations to follow a single community and its associated biological processes. Observations were complemented by a biogeochemical model to address unmeasured nutrient pools and potential grazing–related processes. This study provides a deeper understanding of the relationship between eukaryotic community and productivity in a bloom by (1) coupling direct measurements of primary productivity rates with high resolution metabarcoding techniques for community analysis and (2) comparing metabarcoding to historical methods of community analysis.

## 2 Methods

### 2.1 Bloom simulation and sampling

24–L microcosm bloom simulation experiments were carried out in August 2021 in the Chesapeake Bay aboard the R/V Hugh Sharp. Surface water (2–5 m) was collected from the study site (37.27º N, 76.09º W), located near the mouth of the bay. All materials used to set up and sample the microcosms were cleaned with 1 M HCl and rinsed with MilliQ water. Incubation medium was prepared by pumping surface water (~5 m) directly from the sample site through a series of nylon mesh and glass fiber filters, ending with a 0.3 $\mu$m filter, using a double diaphragm pump into three 24–L translucent polycarbonate (PC) carboys. Surface water inoculum was collected using a rosette system with 12–L Niskin bottles and a CTD profiler from 2–4 m depth and pre–filtered through 210 $\mu$m nylon mesh before being added to the carboys. Filtered surface medium (21.6 L) and pre–filtered surface water inoculum (2.4 L) were added to each carboy to produce a 10 % inoculation. Additional 0.3 $\mu$m filtered medium and 210 $\mu$m pre–filtered surface water inoculum were collected for DNA analysis. Lastly, $NaNO_3$, $NaH_2PO_4$, and $Na_2SiO_3$ solutions were added to each carboy to achieve final concentrations of approximately 40 $\mu$M, 5 $\mu$M, and 50 $\mu$M, respectively. 40 $\mu$M $NO_3^-$ was chosen to mimic historical observations of nutrient loading in Chesapeake Bay (Harding et al., 2019; Malone et al., 1996) and promote a diatom bloom. A nitrogen:silica:phosphorus ratio of ~10:10:1 is compatible with ambient nutrient ratios observed in lower Chesapeake Bay (Fisher et al., 1992) and with diatom nutrient quotas (Brzezinski, 1985; Lomas et al., 2019). Carboys were incubated for eight days without further nutrient additions in an on–deck water bath, using a seawater flow–through system drawn from surface water and a plastic screen shade covering to keep incubation temperature and light similar to in situ conditions. Continuous light and temperature measurements were recorded between 18:00 on day 1 and 18:00 on day 7 using two Onset HOBO Pendant Temperature/Light data loggers suspended ~10 cm below the surface of the on–deck water bath.

Samples for nutrient concentrations were collected from each carboy three times per day, at approximately 06:00, 12:00, and 18:00 local time. Samples were filtered through a 0.22 $\mu$m syringe filter into 50–mL plastic conical tubes and stored at –20°C until analysis.

Samples for pigment analysis were collected twice daily at 12:00 and 18:00 starting on day 2. Duplicate samples (100–400 mL) were filtered onto pre–combusted (500°C for ~5 h) 0.3 $\mu$m 25 mm GF–75 filters using pigment–dedicated filter–holders. Pigment samples were not collected from carboys prior to day 2 due to low biomass concentrations and sample volume restrictions. Instead, two pseudo day 0 samples were produced by filtering ~1 L each of a 10 % inoculum dilution in 0.3 $\mu$m filtered medium immediately after inoculation and without receiving any additional nutrients. Filters were stored individually at -80°C.

DNA samples were taken from each carboy concurrently with the 12:00 pigment and nutrient samples. Three samples were collected in duplicate for carboy C (on days surrounding the expected bloom transition and peak) and two DNA samples were collected from the surface water inoculum. Sample water was collected from each carboy and filtered onto 0.22 $\mu$m Sterivex filters (320–1290 mL of sample water per filter) using a peristaltic pump. Filters were flash–frozen in liquid nitrogen and stored at -80°C until DNA extraction. For more detail on sampling protocols, see **Supporting Information, Methodology**.

## 2.2 Nutrient and pigment concentrations

Reactive nitrite ($NO_2^-$) and silicate (silicic acid, $H_4SiO_4$) concentrations were measured on a ThermoScience Genesys150 UV–Vis spectrophotometer using colorimetric methods (sulfanilamide + *N*–(1–naphthyl)–ethylenediamine and metol sulfite, respectively) modified from Strickland and Parsons (1972). $NO_2^-$ + $NO_3^-$ concentration was measured via chemiluminescent detection using a Teledyne NOx analyzer (NOxBox) according to Braman & Hendrix (1989). $NO_3^-$ concentrations were calculated by subtracting the colorimetrically derived $NO_2^-$ concentrations from their paired $NO_2^-$ + $NO_3^-$ NOxBox concentrations. Detection limits and measurement precision are listed in **Table S1**.

Pigment samples were analyzed by High Performance Liquid Chromatography (HPLC) (Pinckney et al., 1996, 2001). Effective detection limits are listed in **Table S2** and detailed methodology is available in Hooker et al. (2010).

## 2.3 $^{15}$N and $^{13}$C labeled sub–incubations and isotopic analysis for nutrient uptake measurements

Sub–incubations to measure nitrogen and carbon uptake rates using $^{15}$N–$NO_3^-$ and $^{13}$C–$HCO_3^-$ were carried out once per day. Sample water (150–200 mL) was aliquoted into PC bottles for triplicate sub–incubations for each carboy (9 sub–incubations per tracer per day). $^{15}$N–$NaNO_3^-$ (1.5–2.67 mL of a 0.3 mM solution) and $^{13}$C–$NaHCO_3^-$ (1–1.3 mL of a 30 mM solution) were

added to each PC bottle to attain isotopic enrichments of ~8–93 % and ~10 %, respectively, and gently inverted 5 times to mix.

Bottles were then placed into mesh bags to simulate surface water light intensities and incubated for 4 h (approximately 10:00 to 14:00 local time) in a similar water bath system as the carboys. Sub–incubations were terminated by filtration onto pre–combusted 0.3 $\mu$m 25 mm GF–75 filters using label–specific filter–holders, followed by a 2.5 mL 0.3 $\mu$m–filtered seawater rinse (from original medium collection). Filters were stored individually at -20°C until analysis.

## 2.4 POM concentrations and uptake rate measurements

POM filters were fume acidified for 4–6 h in a desiccator with concentrated HCl, packed into tin capsules, and measured using a Sercon ANCA–SL Elemental Analyzer and a Europa 20/20 Isotope Ratio Mass Spectrometer (EA–IRMS). Standards were prepared from a stock solution of urea and measurements were calibrated using an aminocaproic acid (ACA) standard. All detection limits and measurement precisions are listed in **Table S1**. Transport rates ($\rho$) were calculated according to equations modified from Dugdale and Goering (1967):

$$At\%C = \frac{mol\ ^{13}C}{mol\ ^{13}C + mol\ ^{12}C} \times 100\ \% \tag{1}$$

$$At\%N = \frac{mol\ ^{15}N}{mol\ ^{15}N + mol\ ^{14}N} \times 100\ \% \tag{2}$$

$$C\ transport\ [\mu M\ C\ day^{-1}] = \rho_{HCO_3^-} = \frac{At\%\ POC_{exp} - At\%\ POC_{baseline}}{At\%\ DIC_{exp} - At\%\ DIC_{natural}} \times \frac{[POC]}{hr} \times \frac{12\ light\ hrs}{day} \tag{3}$$

$$N\ transport\ [\mu M\ N\ day^{-1}] = \rho_{NO_3^-} = \frac{At\%\ PON_{exp} - At\%\ PON_{baseline}}{At\%\ DIN_{exp} - At\%\ N_{air}} \times \frac{[PON]}{hr} \times \frac{12\ light\ hrs}{day} \tag{4}$$

At%C and At%N are the atom percent for carbon and nitrogen terms and [POC] and [PON] are in units of $\mu$M C and $\mu$M N, respectively. Because $^{13}$C and $^{15}$N labeled sub–incubations were done separately and concurrently each day, it was possible to compare paired labeled–unlabeled incubations, i.e., one set of sub–incubations received either $^{13}$C or $^{15}$N and the other did not. Therefore, At%POM$_{exp}$ - At%POM$_{baseline}$ was used instead of At%POM$_{final}$ - At%POM$_{initial}$ for transport rate calculations. At%POC$_{exp}$ is the "experimental" At%POC (i.e., the At%POC of the incubations enriched with $^{13}$C tracer) and At%POC$_{baseline}$

is the At%POC of the incubations which received $^{15}$N tracer instead of $^{13}$C. The analogous comparison was used for At%PON$_{exp}$ and At%PON$_{baseline}$. A constant DIC concentration of 1.8 mM was assumed according to the relationship between DIC concentration and salinity in Chesapeake Bay (Lee et al., 2015) and an average natural abundance $^{13}$C of -25 ‰ (PDB) was used for all C–uptake calculations (Spiker, 1980; Guo et al., 1996). A natural abundance of 0.365 % was used for At%N$_{air}$ and assumed to be the same in the unlabeled DIN. Uptake per hour was converted to uptake per day assuming 12 light hours per

155 day. Specific uptake rates (V) were calculated as:

$$V_{HCO_3^-}\ [day^{-1}] = \frac{\rho_{HCO_3^-}}{[POC]} \tag{5}$$

$$V_{NO_3^-} \ [day^{-1}] = \frac{\rho_{NO_3^-}}{[PON]} \hspace{4cm} (6)$$

## 2.5 Nutrient–Phytoplankton–Zooplankton (NPZ) model

A detailed description of the model is available in the **Supporting Information, Methodology**. Briefly, a simplified version of the biogeochemical model Carbon, Ocean Biogeochemistry and Lower Trophics version 2 (COBALTv2, Stock et al. 2020) was implemented to investigate potential grazing rates and grazing–associated changes in biogeochemical parameters during the bloom simulation incubations. This model includes a single nutrient (nitrate), and biological components are parameterized based on a large open ocean dataset to represent a single phytoplankton community (diatoms) and a single zooplankton community (medium–sized copepods).

The model assumes a mass balance of five nitrogen pools: dissolved inorganic nitrogen (DIN) in the form of nitrate ($N_{NO3}$), particulate organic nitrogen (PON) in the forms of phytoplankton ($N_{Phyto}$), zooplankton ($N_{Zoo}$), and detritus ($N_{Detritus}$), and dissolved organic nitrogen (DON; $N_{DOM}$). The model simulates the temporal evolution of the system assuming no spatial variations. The initial nitrogen pools were determined from the observed day 1 nitrate concentration ($[NO_3^-]_{avg} \approx 45 \ \mu$mol $kg^{-1}$), an estimation of day 1 $N_{Phyto}$ (5 $\mu$mol $kg^{-1}$) from observed day 2 Chl–a concentrations, and an assumed average 1:100 biomass ratio of zooplankton to phytoplankton on day 1. The average starting concentration of $N_{Phyto}$ was chosen to balance an initial phytoplankton population which was large enough to trigger a bloom, but small enough to roughly match day 2 chlorophyll concentrations. The Redfield ratio of 106C:16N (Redfield, 1934) was used to convert PON and DON into their respective carbon (C) pools and a $C_{Phyto}$:Chl–a ratio of 200 was used to convert between POC and Chl–a concentrations. Sensitivity testing was performed by independently and randomly varying the initial $N_{Phyto}$, $N_{Zoo,}$ and $N_{NO3}$ by ± 20 % across 1000 model iterations.

## 2.6 DNA extraction and 18S V4 sequencing

DNA extraction and cleanup was performed using the ZymoBIOMICS DNA/RNA miniprep kit (Zymo Research, Irvine CA). Universal primer set 515F (Caporaso et al., 2011) and 951R (Mangot et al., 2013; Lepère et al., 2016) were used for PCR to amplify the V4 region of the 18S rRNA gene. (See **Supporting Information, Methodology** and **Table S3** for PCR details). The purified amplicons were indexed using an Illumina indexing kit, pooled, and sequenced for paired–end 2x250 bp sequences on an Illumina MiSeq platform at the Princeton University Genomics Core Facility.

## 2.7 18S sequence clustering and quality control

Demultiplexed sequences were trimmed using prinseq v0.20.4 (Schmieder and Edwards, 2011) to remove primers and low quality read ends. Paired–end reads were merged using the Illumina utilities package v2.6 (Eren et al., 2013) with default parameters. Merged pairs were discarded if more than one third of the first half of either read had a Q–score below Q30, in

accordance with Minoche et al. (2011), or if Q–score fell below Q15 for any base in the overlapped region. Merged reads were then dereplicated and potential chimeras were identified using uchime in vsearch v2.10.4 (Edgar et al., 2011; Rognes et al., 2016). Reads were clustered into operational taxonomic units (OTUs) using swarm v3 with a neighbor nucleotide difference of 1 (d=1) (Mahé et al., 2021). Taxonomy was assigned using usearch_global (vsearch v2.10.4) and reference database PR$^2$ v4.14.0 (Guillou et al., 2012). OTUs were removed if: (1) the representative sequence had less than 10 reads, (2) the OTU represented < 0.01 % of total reads in all samples, (3) > 50 % of reads within the OTU were identified as potentially chimeric, or (4) the OTU was < 90 % identical to the associated PR$^2$ reference sequence, resulting in 335,896 reads which passed quality checks. Lastly, OTUs with a < 97 % match to a PR$^2$ reference sequence were re–classified as "Other [Class]" and combined according to class. E.g., the reads from all OTUs with a 90 to < 97% match to a Bacillariophyta sequence in the PR$^2$ database were combined into one OTU classified as "Other Bacillariophyta." All other OTUs with a ≥ 97% match to a reference were combined if their associated PR$^2$ reference sequence was the same, resulting in 150 total OTUs.

## 2.8 Relative abundance, alpha, and beta diversity analyses

OTUs remaining after quality control were further manipulated for downstream analysis. For relative abundance and alpha diversity analyses (OTU richness and Shannon index), carboy C duplicates were combined by day and the two inoculum samples were combined and used as "Day 0" for all carboys. Lastly, all OTUs identified by PR$^2$ as metazoan (4 OTUs after combination; 42.4 % of all sequences) and bacterial (7 OTUs after combination; 10.4 % of all sequences) were removed for all relative abundance and diversity analyses.

Diversity analysis was performed using the phyloseq v1.44.0 (McMurdie and Holmes, 2013) and microViz v0.12.1 (Barnett et al., 2021) R packages. The centered log–ratio (clr) transformed Aitchison distance was used to calculate beta diversity. A Canonical Analysis of Principal coordinates (CAP) plot was used to compare beta diversity with observations of environmental parameters using the vegan v2.6-6 package in R (Oksanen et al., 2024).

## 2.9 Statistical analysis

Statistical analyses were performed using R v4.3.1 (R Core Team, 2023) and the MATLAB Statistic and Machine Learning Toolbox v24.1 (MathWorks Inc., 2024). In order to assess the significance of the daily variability of rate measurements and alpha diversity, as well as the fit of PDR regression curves, the car v3.1.3 (Fox and Weisberg, 2019) package in R was used to check if residuals were normally distributed using both a Shapiro–Wilk test (significant p–value < 0.01) and a visual inspection of a quantile–quantile plot. Normally distributed data were compared using Pearson correlation or a one–way ANOVA with a Tukey HSD family–wise comparison, and non–normally distributed data were compared using Spearman correlation or a Kruskal–Wallis test with Bonferroni adjusted pairwise comparisons (significant p–value < 0.05). Beta diversity clustering was assessed using homogeneity of dispersion and PERMANOVA tests (significant p–value < 0.05).

## 3 Results

### 3.1 Biogeochemical signatures indicate a diatom bloom

Light (photosynthetically active radiation, PAR) and temperature displayed typical diel cycles during the incubations, ranging 22.5–29 °C with average daily temperature$_{max}$ = 28.0 °C and average daily PAR$_{max}$ = 77.2 Wm$^{-2}$ (median 59.3 Wm$^{-2}$) (**Fig. S1**). Daytime PAR was lowest on day 3 and reached a maximum of 242.8 Wm$^{-2}$ at 14:00 on day 7. Temporal patterns of particulate organic matter (POM), ambient nutrient, and pigment concentrations in all three carboys indicated a phytoplankton bloom that peaked on day 5 (**Fig. 1, S2**). Particulate organic carbon (POC) and nitrogen (PON) remained low in all carboys until day 4

(avg. [POC$_{min}$] = 22.8 ± 4.5 $\mu$M and avg. [PON$_{min}$] = 1.8 ± 1.3 $\mu$M) (**Fig. 1a, S2a–c**). POM then rapidly increased between days 4 and 5, remaining high between days 5 and 7 and reaching average maximum concentrations of 232.0 ± 18.1 $\mu$M ([POC$_{max}$]) and 24.5 ± 4.3 $\mu$M ([PON$_{max}$]).

The increases in POM through day 5 were mirrored by nutrient drawdown, with all three carboys following similar temporal

trends. Nitrate (NO$_3^-$) and silicate (SiO$_4^{4-}$) were supplemented initially to average starting concentrations of 42.6 ± 1.0 $\mu$M and 46.9 ± 1.0 $\mu$M, respectively, and remained high until day 4 in all carboys (**Fig. 1b, S2d–f**). Between noon day 4 and 18:00 day 5, [NO$_3^-$] and [SiO$_4^{4-}$] were rapidly drawn down to 1.1 ± 0.2 $\mu$M and 1.0 ± 0.2 $\mu$M, respectively, and remained low for the remainder of the incubations. Phosphate (PO$_4^{3-}$) concentrations decreased more consistently between days 1 and 5 from an initial average of 4.0 ± 0.4 $\mu$M to 0.6 ± 0.1 $\mu$M by 18:00 day 5 (**Fig. S3**). Nitrite (NO$_2^-$) remained low throughout the bloom,

increasing slightly on day 4 to an average [NO$_2^-$$_{max}$] of 1.2 ± 0.06 $\mu$M on day 5, possibly due to incomplete NO$_3^-$ assimilation by phytoplankton (Collos, 1998; Lomas and Lipschultz, 2006). The small [NO$_2^-$] peak was then quickly depleted following the exhaustion of [NO$_3^-$]. A mass balance of the measured nitrogen species revealed that total measured nitrogen ([NO$_2^-$] + [NO$_3^-$] + PON) was ~45–50 $\mu$M leading up to the bloom peak but was < 30 $\mu$M afterwards. This "gap" in the total measured N may be attributable to unmeasured N pools, such as ammonium or dissolved organic nitrogen (DON) and is addressed in

the NPZ model section below (see also **Fig. S4g–i**).

Chlorophyll a (Chl–a) concentrations also reached their maximum at noon on day 5 for all carboys (avg. Chl–a$_{max}$ = 15.0 ± 2.1 $\mu$g L$^{-1}$) (**Fig. 1c, S2g–i**). Concentrations of fucoxanthin, a diagnostic diatom pigment, followed the same pattern as Chl–a, while peridinin (dinoflagellates) and zeaxanthin (cyanobacteria) remained low, and divinyl chlorophyll a (cyanobacteria)

remained below detection throughout the bloom. Chl–a and fucoxanthin concentrations both decreased following day 5, signifying the decline of the phytoplankton population. Pigment accumulation trends along with the concurrent consumption of SiO$_4^{4-}$ indicate that a diatom bloom occurred, and the late–bloom decrease in Chl–a:chlorophyll c may indicate a shift in phytoplankton community following the bloom peak (Dursun et al., 2021). Additionally, the decoupling between Chl–a and POM concentrations in the late bloom resulted in a large range of POC:Chl–a ratios. POC:Chl–a reached a minimum during

the peak bloom with an average value of 58.6 ± 8.2, which is within the typical range of ~10–200 for large phytoplankton

(e.g.; Laws and Bannister 1980; Schoemann et al. 2005 and references therein). Late bloom ratios were much higher, reaching an average POC:Chl–a > 1000 on day 7, with maximum values of up to 1258.6 ± 247.2 in carboy B (**Fig. S4d–f**), suggesting the accumulation of non–phytoplankton biomass in the late bloom and a potential shift in phytoplankton POC:Chl–a.

### 3.2 Temporal patterns of nutrient transport and specific uptake rates

Temporal patterns of absolute C and N transport rates ($\rho_{HCO3-}$, $\rho_{NO3-}$) were consistent across all carboys. Transport rates remained low prior to day 4 (**Fig. 2**). They then increased slightly on day 4 and reached maximum rates on day 5 with average rates for the three carboys of 186.0 ± 64.7 $\mu$M C d$^{-1}$ and 27.1 ± 8.95 $\mu$M N d$^{-1}$, before quickly dropping again. Days 6 and 7 had transport rates similar to those measured on day 4. The timing of maximum transport rates coincided with the day 5 peak observed in POM and Chl–a concentrations and the highest rates of nutrient depletion. Despite similar patterns, day 5 $\rho_{HCO3-}$

varied between carboys. Peak $\rho_{HCO3-}$ was highest in Carboy C (229.3 ± 80.6 $\mu$M C d$^{-1}$) and lowest in Carboy A (141.6 ± 31.8 $\mu$M C d$^{-1}$) (**Table S4**). However, peak $\rho_{HCO3-}$ were not statistically different between carboys (one–way ANOVA, $p = 0.28$). $\rho_{NO3-}$ was less variable between carboys than between triplicate measurements within a single carboy (**Table S4**).

Biomass specific uptake rates for both C and N ($V_{HCO3-}$, $V_{NO3-}$) peaked on different days, but both displayed a significant

(Kruskal–Wallis, $p = 6.17\times10^{-4}$ and $p = 6.02\times10^{-3}$, respectively) increase between days 3 and 4, one day prior to the increase in absolute transport rates (**Fig. 2, S5**). $V_{HCO3-}$ and $V_{NO3-}$ then remained high until nutrients were depleted, decreasing on day 6. $V_{HCO3-}$ peaked on day 4 with an average rate for all three carboys of 1.0 ± 0.06 day$^{-1}$, and $V_{NO3-}$ peaked on day 5 with an average rate of 1.1 ± 0.12 day$^{-1}$ (**Table S4, Fig. S5**).

### 3.3 NPZ model supports late bloom grazing

A nutrient–phytoplankton–zooplankton (NPZ) model was employed to investigate the "gap" in total measured N and the potential contribution of unmeasured biogeochemical parameters. The model was able to accurately simulate the observed magnitude and temporal evolution of [NO$_3^-$], Chl–a, and POM when including grazing by zooplankton, with the best fit to [NO$_3^-$] (**Fig. 3a–d**). The model slightly overestimated POM during the early bloom, but matched observations from the peak bloom onward. The modeled grazing rate reached an average maximum of 27.3 $\mu$M N d$^{-1}$ and a combined non–phytoplankton

PON ($N_{Zoo}$ + $N_{Detritus}$) which represented 67.5 % of total modeled PON at the end of the bloom. Modeled grazing and respiration rates resulted in a ~20 $\mu$M N accumulation of DON on day 7, which is comparable to the 20–30 $\mu$M N "gap" in the observed N mass balance (**Fig. S4g–i**).

### 3.4 Community succession resulted in distinct temporally varying assemblages

Changes in the eukaryotic community composition were detected during the bloom using amplicon sequencing of the

hypervariable V4 region of the 18S rRNA gene. Swarm clustering (Mahé et al., 2021) and quality control resulted in 143

eukaryotic OTUs. While Metazoa such as copepods were present in the 18S sequences (**Fig. S6**) and a potentially important portion of eukaryotic community, the multicellular nature of Metazoa made it difficult to interpret their relative abundances. Therefore, they were removed from community analyses, leaving 139 non–metazoan eukaryotic OTUs for further analysis.

OTU relative abundances in all three carboys showed succession of the bloom community, with the development of three distinct, temporally driven assemblages: early– (days 2 and 3), mid– (days 4 and 5) and late–bloom (days 6 and 7) (metadata summarized in **Table S5**). Community composition in early–bloom samples was similar to inoculum samples, with the exception of parasitic Syndiniales OTUs. Syndiniales OTUs had a high relative abundance in inoculum samples but remained low in carboy samples (**Fig. 4a–c**). Most early bloom sequences were from the green algae Mamiellophyceae, diatoms 290 (Bacillariophyta), dinoflagellates (Dinophyceae), unclassified marine Stramenopiles (MAST–1), and heterotrophic Spirotrichea OTUs. By the mid– and late–bloom, relative abundances indicated that the community was composed primarily of diatoms and dinoflagellates, though the ratio of diatom and dinoflagellate relative abundances varied between carboys. At the class level, the mid– and late–bloom assemblages are distinguished by a late–bloom increase in the relative abundance of heterotrophic organisms like Filosa–Thecofilosea and, in Carboy C, Spirotrichea OTUs. Their increase is consistent with 295 grazing and an accumulation of non–phytoplankton material in the late–bloom.

The early–, mid–, and late–bloom community shifts resulted in three distinct clusters in the beta diversity analysis (PERMANOVA, $p = 0.001$, $R^2 = 0.60$) (**Fig. 4g**). Early–bloom samples cluster with inoculum samples and late–bloom samples cluster together regardless of carboy. Mid–bloom samples show the greatest variability between both carboys and day, except 300 for carboy C, which had relatively similar communities on days 4 and 5. Additionally, variability between biological duplicates (inoculum and carboy C, days 3, 5 and 6) is of the same magnitude as variability between samples from different carboys on the same day (**Fig. 4g**). Both day and [$NO_3^-$] vary along CAP1. Therefore, CAP1, which explains 42.2 % of community composition, likely corresponds to a combination of time since inoculation and nutrient availability.

The alpha diversity of the non–metazoan eukaryotic community peaked during the early–bloom, decreased during the mid–bloom, and did not significantly change between the mid– and late–bloom. This pattern was most prominent in the OTU richness (number of OTUs), which separated into two statistically distinct groups: the high diversity early–bloom and the low diversity mid/late–bloom (**Fig. S7a**). Conversely, the Shannon index (H) did not significantly vary throughout the bloom despite sharing the same temporal trend (**Fig. 4d–f, S7b**). H ranged from 0.5–3.4 with an average of 2.5. While H decreased 310 slightly in carboys A and C during the bloom peak, it did not drop below 2. The mid–bloom decrease in H was likely driven by carboy B's ~40 % decline in H from an early–bloom average of 3.02 to a mid/late–bloom average of 1.24. Overall, while the alpha diversity did decrease leading up to the bloom peak, the peak was not characterized by a significant or consistent diversity minimum.

### 3.5 High diversity and succession of the diatom bloom assemblage

A more comprehensive analysis of the diatom assemblage was performed because of their prominence among the phytoplankton community and because the biogeochemical data indicated their dominance during the peak bloom. The relative abundance of diatom sequences generally increased through the early–bloom, reached a maximum of up to ~60 % (in carboy C) during the mid–bloom, and then decreased during the late–bloom. The relative abundance of diatoms in carboy B peaked slightly earlier and had lower relative abundances than the other carboys during the mid– and late–bloom, except for a day 6

peak of an araphid pennate diatom OTU (**Fig. 5a–c**) identified as a *Thalassionema sp.* via BLAST search. Succession was also evident in the composition of the diatom assemblage. The highest relative abundance diatom sequences were all cosmopolitan; *Chaetoceros* dominated the early–bloom, *Thalassiosira* the mid–bloom, and *Thalassionema* the late–bloom. *Ditylum* sequences were also highly abundant, especially just preceding the peak bloom. The relative abundances of most diatom genera were dominated by a single OTU, except for *Chaetoceros*, which had > 10 OTUs that were relatively evenly represented.


Diatom–specific alpha diversity was more consistent between carboys and displayed clearer temporal trends compared to overall community alpha diversity. Diatom alpha diversity increased during the early–bloom and decreased during the mid–bloom, with variable trends during the late–bloom. However, diatom richness never dropped below 17 in any carboy (**Fig. 5g–i**) and dozens of diatom OTUs, spanning multiple genera, were present at the peak of the bloom.

## 4 Discussion

### 4.1 Diatom–driven peak bloom patterns

Maximum nutrient transport rates, Chl–a accumulation, and nutrient drawdown all occurred on day 5 and indicate that a diatom bloom occurred (**Fig. 6**). While seasonal succession of the phytoplankton community is well documented in Chesapeake Bay and predicts a highly diverse community of diatoms, dinoflagellates, and cyanobacteria in the summer (Adolf et al., 2006;

Harding et al., 2015; Marshall et al., 2005), the pigment data did not indicate that dinoflagellates or cyanobacteria contributed significantly to the biomass (**Fig. 1c, S2 g–i**). Light patterns could not explain the trend in POC:Chl–a observed in the carboy incubations (**Fig. S1**). Instead, high nutrient conditions (Laws and Bannister, 1980; Schoemann et al., 2005; Arteaga et al., 2016) and diatom–dominated biomass (Tada et al., 2000; Sathyendranath et al., 2009) resulted in low total community POC:Chl–a during the mid–bloom.


The observed maximum C and N transport rates in this study were higher than previously reported summer rates in the lower bay (e.g.; Sellner 1983; Glibert et al. 1995; Marshall and Nesius 1996; Bradley et al. 2010). $\rho_{NO3-}$ in particular was often higher than previous observations of total dissolved N transport ($\rho_{TDN}$, the sum of uptake of all forms of N) (e.g.; Glibert et al. 1995; Bradley et al. 2010). Similarly, maximum Chl–a–normalized C transport ($V_{C\_Chla}$, **Fig. S8**) and $V_{NO3-}$ rates were also higher

than previous reports (Adolf et al., 2006; Bradley et al., 2010) (additional information in the **Supporting Information, Analysis**). High nutrient transport rates and a stronger $NO_3^-$ preference are all associated with diatom blooms, especially in the lower bay (Glibert et al., 1995; Marshall and Nesius, 1996; Sellner, 1983). Additionally, high summer $V_{C\_Chla}$ is positively correlated to the fraction of Chl–a attributable to diatoms (Adolf et al., 2006) and $V_{NO3^-}$ tends to be higher in larger (> 35um) size fractions than total community (Bradley et al., 2010).

Although specific uptake rates were high compared to previous Chesapeake Bay measurements, they were similar to those previously observed in a bloom mesocosm (Fawcett and Ward, 2011) and the open ocean (Van Oostende et al., 2017). Both $V_{HCO3^-}$ and $V_{NO3^-}$ significantly increased one day prior to the bloom peak (**Fig S5**). The observed increase in specific uptake rates prior to absolute transport rates is consistent with evidence that blooming phytoplankton (e.g., diatoms) are able to exploit nutrient–rich conditions by both increasing and maintaining specific uptake rates. This ability of diatoms to shift–up their nutrient uptake in response to a sudden nutrient addition is a well–documented phenomenon (e.g.; Kudela and Dugdale 2000; Fawcett and Ward 2011; Lampe et al. 2018), including in Chesapeake Bay (Malone et al., 1996).

Both absolute transport and specific uptake rates may be overestimated after day 5, as ambient $[NO_3^-]$ became depleted. However, it is unlikely that temporal patterns observed in the transport or specific uptake rates were altered by these factors (additional information in the **Supporting Information, Analysis** and **Table S6**).

The depletion of nutrients by noon on day 5 and high nutrient requirements for large blooming phytoplankton suggests that the diatom community was unable to sustain growth, leading to the bloom's demise following day 5. We observed a strong coupling between $NO_3^-$ and $SiO_4^{4-}$ drawdown rates (**Fig. S9c**), which matched average diatom N:Si quotas, between days 2–5 (Brzezinski, 1985; Fisher et al., 1992). However, following the combined kinetics– and stoichiometry–based thresholds outlined in Liang et al. (2019), nutrients were only transiently limiting during the latter half of the bloom (**Table S7**). There were two timepoints when nutrients were limited in more than one carboy: $SiO_4^{4-}$ on the evening of day 5 and $NO_3^-$ on the evening of day 6, however, neither of these nutrients were limited in back–to–back samples. Additional analysis of the NPZ model revealed that nutrients likely became partially limiting following the bloom peak. Modeled nitrate limitation did not match the timing of the threshold–based limitation, but did reach an average maximum of ~60 % at the end of the incubations (**Fig. 3b**). The combination of incomplete or transient nutrient limitation and consistently high POM during the late–bloom indicate that factors other than nutrient availability likely contributed to the bloom decline; the potential role of grazing is further investigated below.

## 4.2 Late–bloom patterns were influenced by grazing

In addition to or instead of nutrient limitation, the bloom decline may have been due to top–down factors such as grazing, parasitism, and viral infection. Biogeochemical measurements, an NPZ model, and 18S community analysis provide evidence for grazing.

In contrast to the low peak bloom POC:Chl–a, the extreme POC:Chl–a values observed at the end of the carboy incubations far exceeded both the average phytoplankton POC:Chl–a of ~40–90 observed in Chesapeake Bay (Cerco 2000 and the references therein) and previous observations of maximum phytoplankton POC:Chl–a < 500 (e.g.; Laws and Bannister, 1980; Jakobsen and Markager, 2016), with average day 7 values over 10–fold greater than expected for the region. Low available nutrients during the late–bloom likely contributed to an increase in phytoplankton POC:Chl–a (e.g.; Behrenfeld et al., 2005; Arteaga et al., 2016 and the references therein), however an increase in non–phytoplankton biomass was also necessary to explain the ratios > 1000 observed in this study. High total community POC:Chl–a is primarily influenced by the ratio of phytoplankton POC to zooplankton and detrital POC (Banse, 1977). Even though the inoculum was prefiltered using a 210 $\mu$m mesh, metazoan sequences were present in every DNA sample. Therefore, it is likely that smaller Metazoa passed through the initial filter and grew in response to the bloom. Grazing would also contribute to unmeasured N pools, as DON would be released during grazing or parasitic and viral lysis (Bronk, 2002; Park et al., 2004). Given the timing and preparation of the experiments, as well as the absence of Chlorophyllide a (a Chl–a degradation product) in all pigment samples, phytoplankton detritus was unlikely to have contributed significantly to POM.

To further investigate potential role of copepods, we used an NPZ model to simulate the transfer of N to different pools during the bloom, mediated by diatom and copepod processes. The model outputs for $NO_3^-$, Chl–a, and POM matched observations (**Fig. 3a–d**), providing support for the application of the model as a proxy for unmeasured N pools, such as DON, and fluxes between N pools. The NPZ model, despite being a simplification of the full community dynamics, showed that grazer–phytoplankton interactions could explain the majority of observed biogeochemical patterns in the carboy bloom simulations given the initial nutrient conditions. Transfer of PON from phytoplankton into zooplankton biomass and detritus match and explain the observed high PON at the end of the bloom despite a decrease in chlorophyll, and therefore phytoplankton biomass. Additionally, DON released as a direct result of grazing in the model can account for a significant portion of the observed N mass balance "gap" (**Fig. 3f**).

Simplifications in the model prevented over–parameterization, but led to slight deviations between model outputs and observations (additional information in the **Supporting Information, Analysis**). Despite these minor deviations, the consensus between observations and a simplified NPZ model demonstrates that even a diverse diatom assemblage acts in accordance with globally averaged diatom behavior under blooming conditions. Furthermore, the modeled grazing contrasts

with previous studies which found a decoupling of diatoms and copepods in the open ocean (Lima–Mendez et al. 2015) and suggests that the diatom community in a shallow, coastal environment can be controlled both by bottom–up and top–down factors.

### 4.3 Replication of bloom timing and community between carboys

Patterns of 18S–based community composition were similar across carboys. Beta diversity analysis showed that between–carboy dissimilarity was of comparable magnitude to the dissimilarity between biological replicates for carboy C, except on day 4 (**Fig. 4g**). Day 4 is a transitional period between the early–bloom and peak bloom assemblages; all three carboys appear to have been trending toward a similar peak bloom assemblage, but rapid community succession and slight variability in the exact timing of the peak may have caused the observed divergence between assemblages. Carboy C showed a similar assemblage on days 4 and 5, implying that the peak bloom community was present slightly earlier than in other carboys, concurrent with the uptick in specific uptake rates and decrease in alpha diversity (**Fig. 6**). This indicates that the community being established was responsible for the observed shift in nutrient acquisition strategy. The variability in the timing of the bloom peak may be due to minor differences in the starting community that each carboy received, as seen in the dissimilarity present between replicate inoculum samples despite being filtered from the same stock of water (**Fig. 4g**). Regardless of community dissimilarities between inoculum duplicates or between carboys during the mid–bloom, the early– and late–bloom samples clustered very closely by day. Open ocean processes, for example mixing, which allows for nutrient replenishment and the introduction of new organisms, would likely cause closed system experiments to deviate from the natural environment on longer timescales or larger study regions. However, robust, short timescale events like blooms are less likely to be disrupted by mixing. The replicability between incubations may indicate that so long as inoculum volume and percentage are high enough to ensure the presence of the same potential blooming organisms, carboy experiments can be replicable and good approximations of in situ community dynamics.

### 4.4 Disentangling the bloom community: pigment– vs. 18S–based analyses and the role of dinoflagellates

Very few studies have performed 18S–based analyses in the main stem of the Chesapeake Bay (Cram et al., 2024; Wang et al., 2024), but a comparison of microscopy– and 18S–based community analysis revealed a strong correlation between cell counts and relative abundance of major phytoplankton groups (Wang et al., 2024). However, in the current study, the high dinoflagellate 18S relative abundances in the mid– and late–bloom (**Fig. 4a–c**) contradict the diatom–dominated story suggested by the biogeochemical data and thus required additional inspection.

Diatoms often dominate the Chesapeake Bay phytoplankton community, but many studies have documented summer–fall harmful dinoflagellate blooms in the bay (e.g.; Li et al. 2015 and references therein). A large portion of the dinoflagellate sequences in this study belonged to *Karenia mikimotoi* (**Fig. S10**) (previously *Gymnodinium mikimotoi, Gymnodinium* cf. *nagasakiense,* and *Gyrodinium aureolum*) (Hansen et al., 2000; Li et al., 2019). *Karenia spp.* are globally distributed

dinoflagellates associated with harmful algal blooms (e.g.; Glibert et al. 2014; Li et al. 2019). A key feature of *K. mikimotoi* is that it contains fucoxanthin instead of the typical dinoflagellate diagnostic pigment peridinin, making it difficult to distinguish between *K. mikimotoi* and diatoms based on fucoxanthin vs. peridinin alone (Richardson and Pinckney, 2004; Li et al., 2019). However, other *Karenia*–associated accessory pigments were absent during the mid– and late–bloom (**Fig. S11**) and *K. mikimotoi*'s growth strategies make it most suited to lower nutrient environments compared to this study (additional information in the **Supporting Information, Analysis**). Therefore, it is unlikely that *K. mikimotoi* was a consequential component of the biomass. Instead, we suggest that the apparent high relative abundance of dinoflagellates is an issue of 18S gene copy number.

The 18S rRNA gene copy number (GCN) can vary between species and individuals, leading to bias in relative abundance data (Guo et al. 2016; Martin et al. 2022). Both diatom and dinoflagellate 18S GCNs can range over several orders of magnitude, but dinoflagellates on average have almost 30 times as many copies as diatoms (Martin et al., 2022). Corrections to relative abundance data based on GCN studies can improve the accuracy and inform the interpretation of 18S amplicon data, but implementation is limited due to the low number of (and variability between) GCN studies. Even though a previous Chesapeake Bay study showed that relative abundances were correlated to cell counts at a high taxonomic level (Wang et al., 2024), relative abundances impacted by GCN are not necessarily indicative of biomass or community relevance. This study exemplifies the pitfalls of relying solely on 18S relative abundance and reinforces that multiple types of data should be implemented to assess community dynamics.

**4.5 18s–based analyses reveal that high diversity is maintained during the bloom**

The average 18S–based H for the whole community (2.5) was lower than previous non–bloom studies in Chesapeake Bay, which found in situ H values of 4–6 (Cram et al., 2024; Wang et al., 2024), but not as low as expected of a bloom. The canonical phytoplankton productivity–diversity relationship (PDR) is characterized by a unimodal function with maximum diversity found in intermediately productive communities (indicated by cell abundances, biomass, or productivity rates). Minimum diversity, as low as one dominant species or H < 0.1, have been observed and modeled in high productivity phytoplankton communities (e.g.; Irigoien et al. 2004; Vallina et al. 2014). A phytoplankton bloom is expected to follow the same patterns, with lowest diversity during the peak of the bloom and concurrent with maximum productivity.

While the alpha diversity in this study did decrease during or near the peak of the bloom (**Fig. 6, S7**), neither the OTU richness nor H dropped as low as expected according to the traditional bloom framework. Higher than expected OTU richness during the bloom peak was largely due to molecular techniques' ability to detect a larger portion of the community at greater taxonomic resolution than microscopy (López–García et al. 2001; Massana and Pedrós–Alió 2008; Xu et al. 2023). Whole community H did not vary significantly between days, partly because H and other diversity measures which account for evenness tend to weaken the unimodal PDR (Skácelová and Lepš, 2014) and make it difficult to observe a traditional diversity

minimum during maximum productivity. Neither H nor OTU richness were correlated with sequencing depth (Pearson, $p = 0.97$ and $p = 0.62$, respectively); however, H values can be heavily impacted by the variability of 18S rRNA GCN. H accounts for the community evenness by using the relative abundance of OTUs, which is biased due to the orders of magnitude difference in 18S rRNA GCN between taxonomic groups in a whole community analysis.

Even when comparing multiple micro- and mesocosm studies, both the magnitude and temporal trends of whole community H were inconsistent (**Fig. S7c**). However, clearer temporal patterns in alpha diversity, which more closely reflected the canonical unimodal PDR, were revealed when isolating an individual taxonomic group. Diatom–specific H in this study increased rapidly over the first few days of the bloom and decreased during the mid–bloom (**Fig. 5d–i, 6a**). While late–bloom diversity varied carboy–to–carboy, the overall trend matched the decrease observed in other bloom simulation experiments (**Fig. S7d**). The classic peak bloom diversity minimum may be more easily observable in the diatom community because the unimodal PDR has typically been used to describe phytoplankton communities, rather than whole communities, when applied to microbes. Diatom–specific H values were also much closer to previous observations, regardless of whether H was determined by microscopy or 18S, indicating that alpha diversity is directly comparable between these methods so long as it is calculated for groups with similar GCNs.

### 4.6 Effect of regionality and methodology on productivity–diversity relationship

This study's PDR was compared to existing open ocean and mesocosm–specific datasets in three ways: (1) richness vs. POC, (2) H vs. POC, and (3) H vs. productivity rates. These methods were chosen to compare our data to previous PDR studies, as well as investigate the PDR in ways that are more suited to molecular data and use true productivity measurements such as C transport rates. Contrary to the classic unimodal PDR, all three PDRs for the Chesapeake Bay experiments were monotonically negative, though H–based PDRs were not significant (**Fig. 7**).

While a unimodal relationship, either expressed as a unimodal curve or a cloud of points whose upper bound is defined by a unimodal curve (Smith, 2007; Skácelová and Lepš, 2014), may be observed in larger regional studies, it may not apply to local diversity patterns, which can appear monotonic (Rosenzweig, 1992). This can be seen when comparing the combined global and local data as a whole to individual localized experiments (**Fig. 7a,b**). Excluding the much higher 18S–derived diversity of this study, the upper bounds of all microscopy–derived data clearly displayed unimodal patterns and the data fit flatter, though still significant, unimodal curves. Additionally, Irigoien et al. (2004) showed that phytoplankton– or zooplankton–specific diversity was a function of both phytoplankton and zooplankton biomass as separate parameters, making the unimodal PDR curve of combined factors less distinctly defined. Contrastingly, the individual localized micro- and mesocosm studies displayed both positive and negative monotonic PDRs. The issue of regionality may occur because a range of several orders of magnitude in productivity is often needed to detect a true unimodal relationship or because taxa appear more cosmopolitan at small scales (Smith, 2007). The latter effect may be heightened in micro- and mesocosm experiments where diversity is

bounded on both ends by the inoculum community. It is also true for many generalist microbes regardless of study scale and may also explain why unimodal, positive, and negative monotonic relationships are all commonly reported in aquatic microbial PDR studies (Smith, 2007; Graham and Duda, 2011; Skácelová and Lepš, 2014). Aside from unimodal phytoplankton community PDRs, negative monotonic relationships are the most common amongst general observations of natural aquatic

microbial communities (Smith, 2007).

The consistently negative whole community PDR of this study supported the common interpretation that competitive exclusion, in this case driven by blooming diatoms, produce the latter (negative) half of the canonical unimodal PDR. However, examining this study in isolation, it is unclear if the negative relationship resulted from an absence of low productivity

observations or because 18S–based diversity measurements reveal greater than expected diversity in the inoculum community, which should have been limited by the low nutrient in situ conditions.

To address this, we compared our study to another whole community 18S–based H vs. productivity rate study (Wang et al., 2021) (**Fig 7c**). Both studies produced negative PDRs, despite including open ocean samples. The combined 18S–based

datasets resulted in the best fitting PDR which included data from this study (Pearson, $p = 3.55 \times 10^{-13}$), with greater significance than either dataset in isolation. This combined dataset covered both a broad range of productivity rates and regionality, but did not result in a clear unimodal PDR. Therefore, while our data by themselves displayed the best linear fit in the traditional Richness vs. POC PDR, the H vs. productivity rate PDR may be representative of larger regional PDR trends with 18S–based diversity.

**5 Conclusions**

This study provides a more direct approach to PDRs by measuring productivity rates concurrent with DNA sampling. Furthermore, it provides additional complexity by analyzing a more comprehensive eukaryotic community, both by using 18S–based diversity analyses and by including non–phytoplankton taxa, over the time course of the bloom. Overall, carboy microcosms reached maximum diversity leading up to the peak bloom, followed by a day 4 transition in the community and

increase in specific uptake rates, which led to a day 5 peak in Chl–a, fucoxanthin, and transport rates, and finally resulted in a late–bloom reduction in all phytoplankton signatures and POM accumulation (summarized in **Fig. 6**), as well as increased potential grazing rates.

The use of 18S amplicon sequencing to analyze the development and waning of a phytoplankton bloom thus provides a new

lens through which to view the patterns of species succession and relationships between diversity and productivity. Although the 18S data detected a larger portion of the community with more taxonomic resolution than was possible with microscopy or pigments, a general pattern of lower diversity at the height of the bloom was still evident (**Fig. 6**). Diatoms dominated the

bloom, and therefore the patterns of nutrient utilization and biomass production. The characteristic relationships between productivity (at high levels of primary production) and diversity held for the diatom component of the bloom, as well as for

the entire community. The most abundant group (diatoms), however, was represented by multiple genera and multiple "species" within some genera, rather than by one or a few major species. Many other phytoplankton genera, including dinoflagellates, persisted through the bloom, clustering into assemblages that characterized different (early, mid, late) phases of the bloom. It was not possible to attribute the decline of the bloom to one factor, since grazing and nutrient depletion coincided. Nutrient depletion during the late–bloom caused growth rates to decrease and contributed to the bloom's demise,

but late–bloom POM accumulation, high POC:Chl–a ratios, and modeled grazing rates indicate that grazing was also necessary to explain the observed trends. General reproducibility of the temporal patterns of uptake rates and succession in the 24–L microcosms, as well as successful simulation by an oceanic scale model, suggest that these results provide insight into coastal and oceanic phytoplankton bloom dynamics. Understanding these dynamics along transition zones such as the lower bay is increasingly important as rising sea level pushes high salinity conditions and communities, known to be the most responsive

to nutrient additions (Adolf et al., 2006), further up the bay where they will be more susceptible to nutrient loading (Bilkovic et al., 2019).

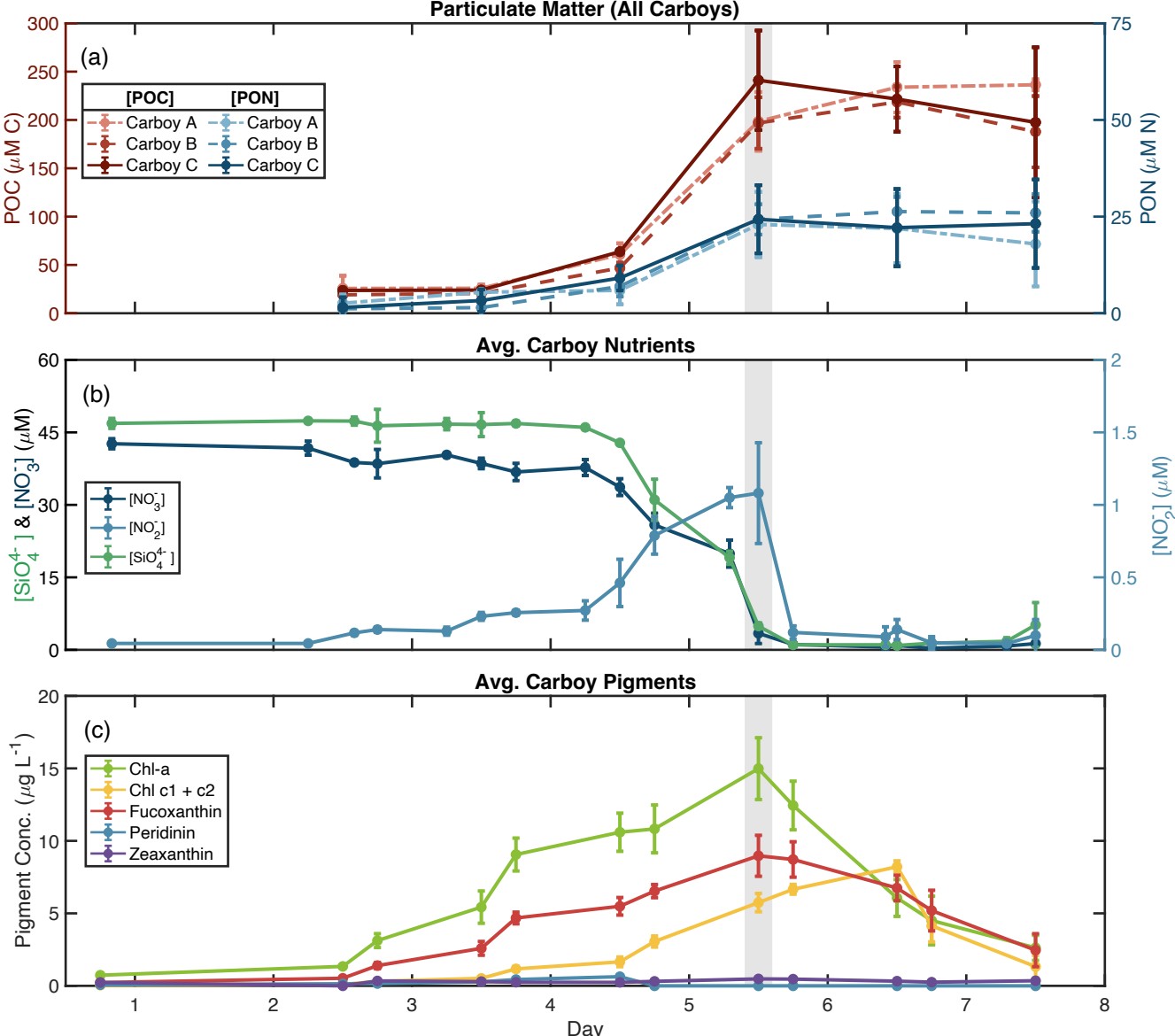

**Figure 1: Biogeochemical data indicate a bloom.** (a) Particulate organic carbon (POC) and nitrogen (PON) concentrations for all carboys over the course of the bloom simulation. POC is in red/pink, PON is in blue, and line type denotes carboy. Error bars in (a) represent the standard deviation of sample replicates. Average carboy concentrations are shown for (b) nutrients and (c) pigments (data for individual carboys are presented in Fig. S2). (b) Nutrient measurements during the bloom simulation are presented as dark blue lines for nitrate ($[NO_3^-]$), light blue lines for nitrite ($[NO_2^-]$), and green lines for silicate ($[SiO_4^{4+}]$). (c) Pigment concentrations for chlorophyll a (Chl-a), chlorophyll c (Chl c1 + c2), and diagnostic pigments for diatoms (fucoxanthin), dinoflagellates (peridinin), and cyanobacteria (zeaxanthin). Error bars in (b, c) represent the standard deviation between carboys. Day 0 pigment concentrations are from a separate 10 % inoculation "dilution" (described in the methodology), rather than individual carboys. Time is shown as days since carboy inoculation where day 1 begins at 00:00 following inoculation. The grey shaded region indicates the peak bloom (~noon day 5).

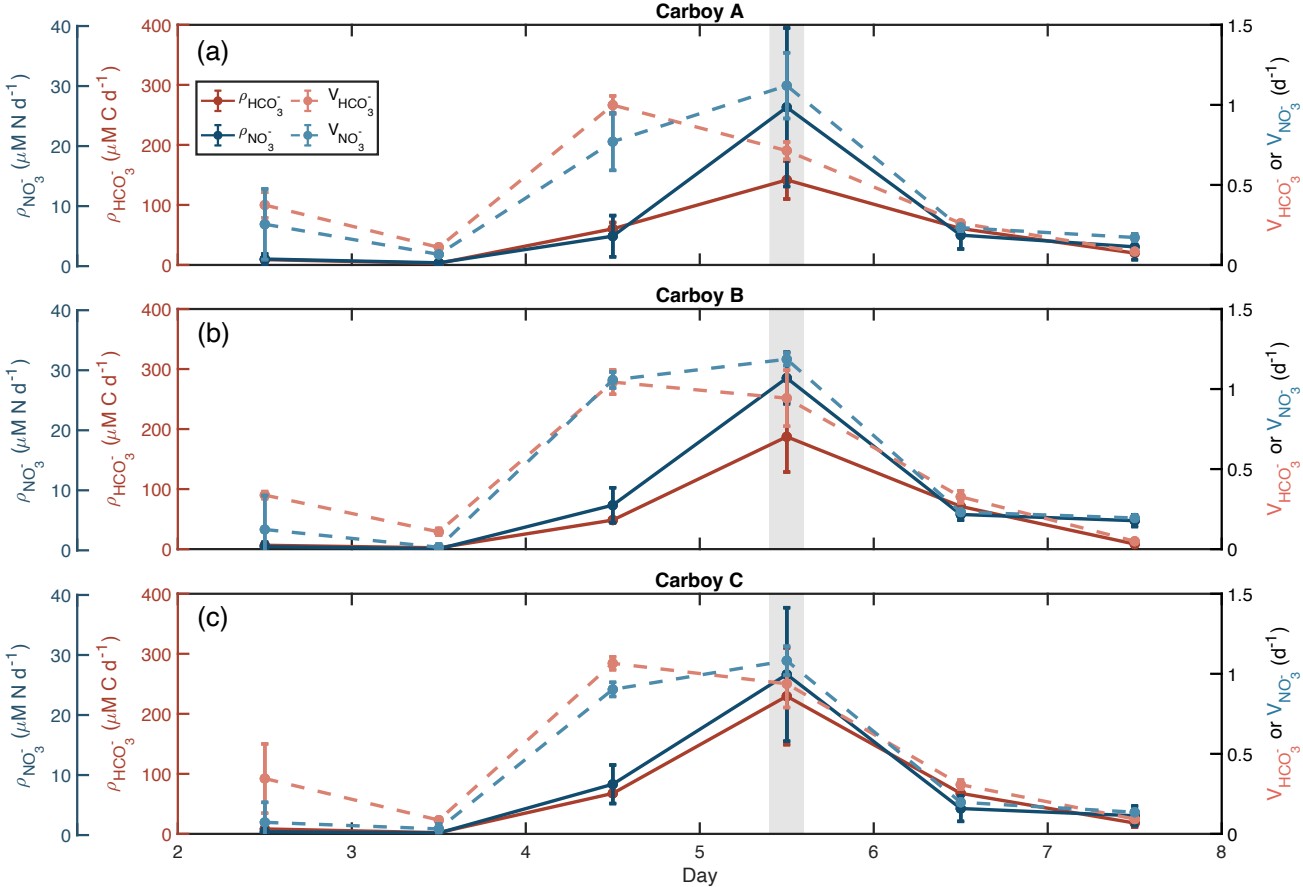

**Figure 2: Specific uptake rates sharply increase before the bloom peak.** Carbon (C) and nitrogen (N) uptake rates are shown in red/pink and blue, respectively for each carboy. Dark solid lines are absolute uptake rates ($\rho$) calculated from labeled $^{15}N$–$NO_3^-$ and $^{13}C$–$HCO_3^-$ sub-incubation experiments. Light dashed lines are specific uptake rates (V), which are equal to $\rho$ normalized to POC for $V_{HCO_3^-}$ and PON for $V_{NO_3^-}$. Error bars show standard deviation of triplicate samples. The light grey shaded region indicates the peak bloom (~noon day 5).

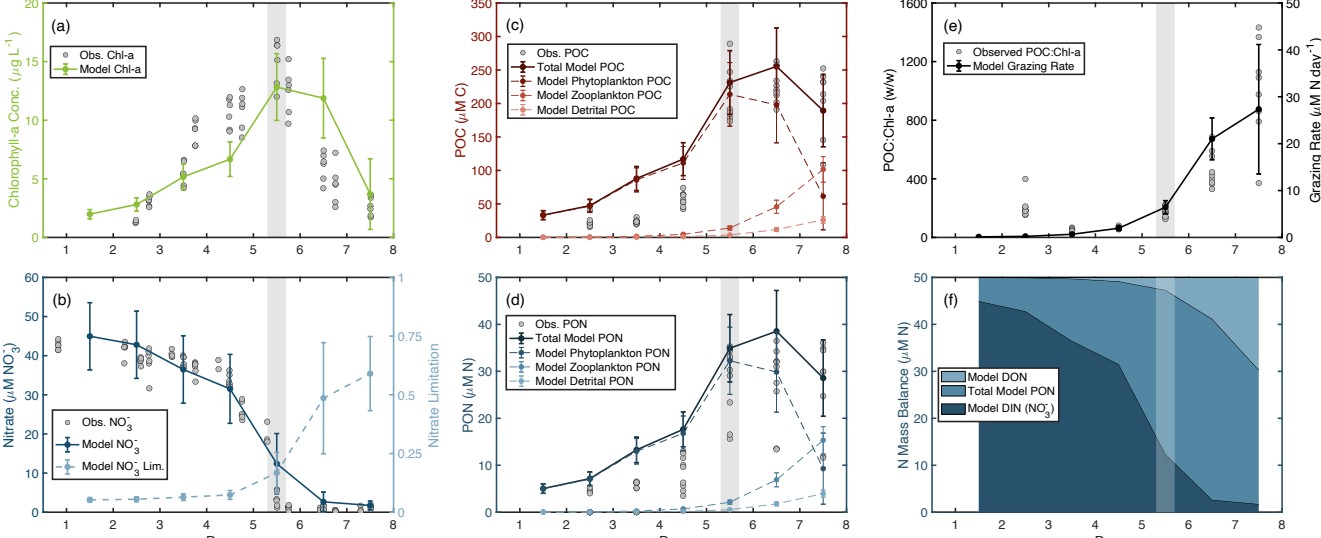

**Figure 3: NPZ model outputs indicate potential grazing.** Model outputs are plotted with observations for (a) chlorophyll a, (b) nitrate concentration ($[NO_3^-]$), (c) particulate organic carbon (POC), and (d) particulate organic nitrogen (PON) concentrations. (b) Model–derived nitrate limitation ($1-N_{lim}$, eq. S6) is plotted alongside $[NO_3^-]$, such that 0 indicates no growth limitation due to nitrate and 1 indicates complete limitation of growth due to nitrate. Potential grazing rates are represented by (e) the modeled grazing rates and compared to observations of POC:Chl–a. And (f) the nitrogen mass balance shows the relative contribution of various nitrogen pools to the total nitrogen budget of the modeled bloom. Observed data are shown as grey dots. The grey shaded area indicates the peak bloom. Modeled data shown as lines with error bars representing the standard deviation of sensitivity testing.

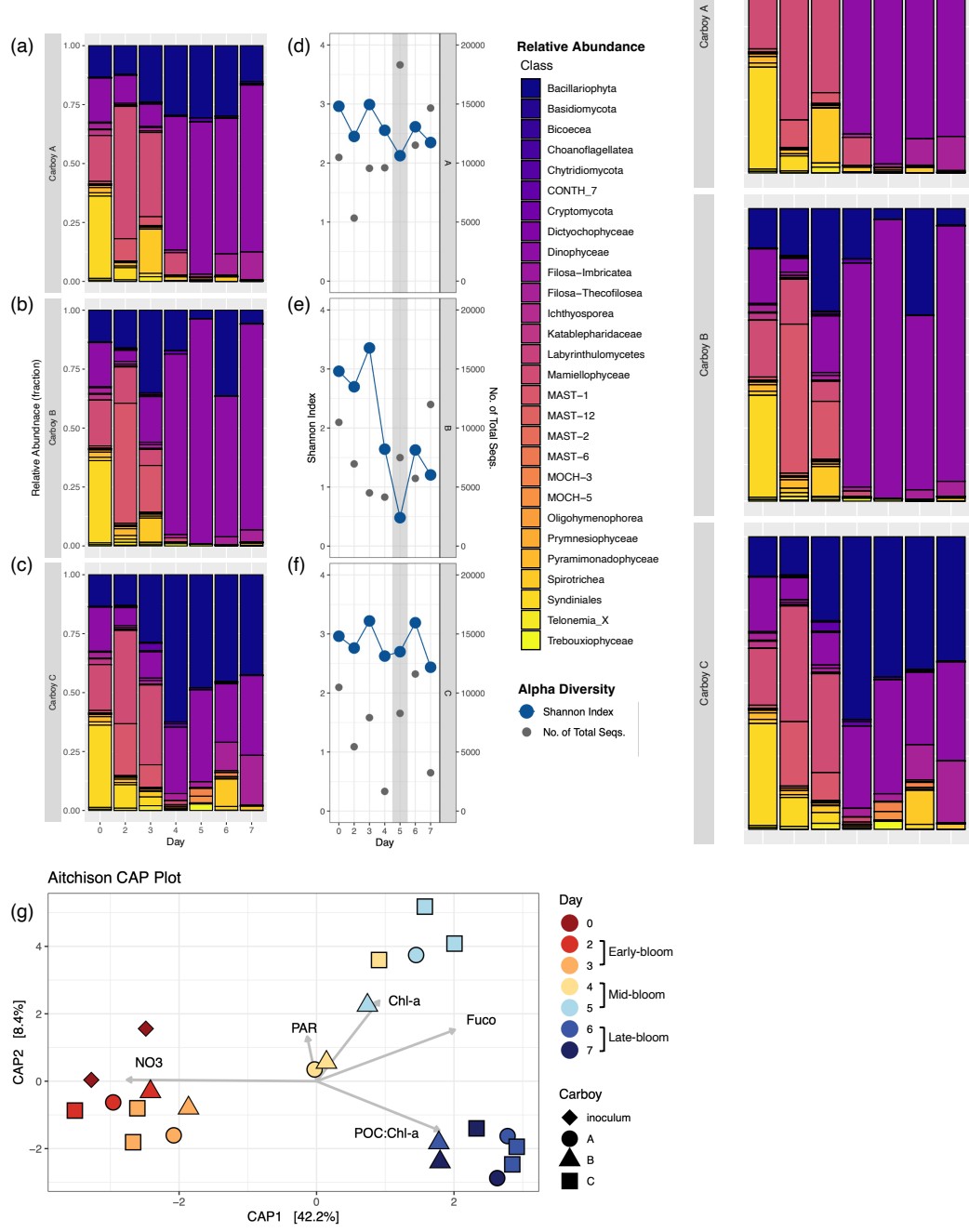

**Figure 4: Community succession and diversity during the bloom.** (a–c) Relative abundance shown as the fraction of non–metazoan 18S–derived eukaryotic OTUs for each carboy, colored by taxonomic class. Sequence counts for biological duplicates of inoculum and carboy C samples were merged before analysis. "Day" corresponds to the number of days since inoculation, with day 0 for each carboy represented by the same merged inoculum sample. (e–f) Shannon alpha diversity (blue) and the number of non–metazoan sequences (grey) on the same "Day" scale. The grey shaded region indicates the peak bloom. (g) Aitchison distance beta diversity measures. Color corresponds to the number of days since inoculation and shape corresponds to carboy. Beta diversity is plotted alongside relevant environmental parameters: NO3 (nitrate concentration), PAR (photosynthetically active radiation), Chl-a (chlorophyll a concentration), Fuco (fucoxanthin concentration), and POC:Chl-a (carbon–to–chlorophyll a ratio).

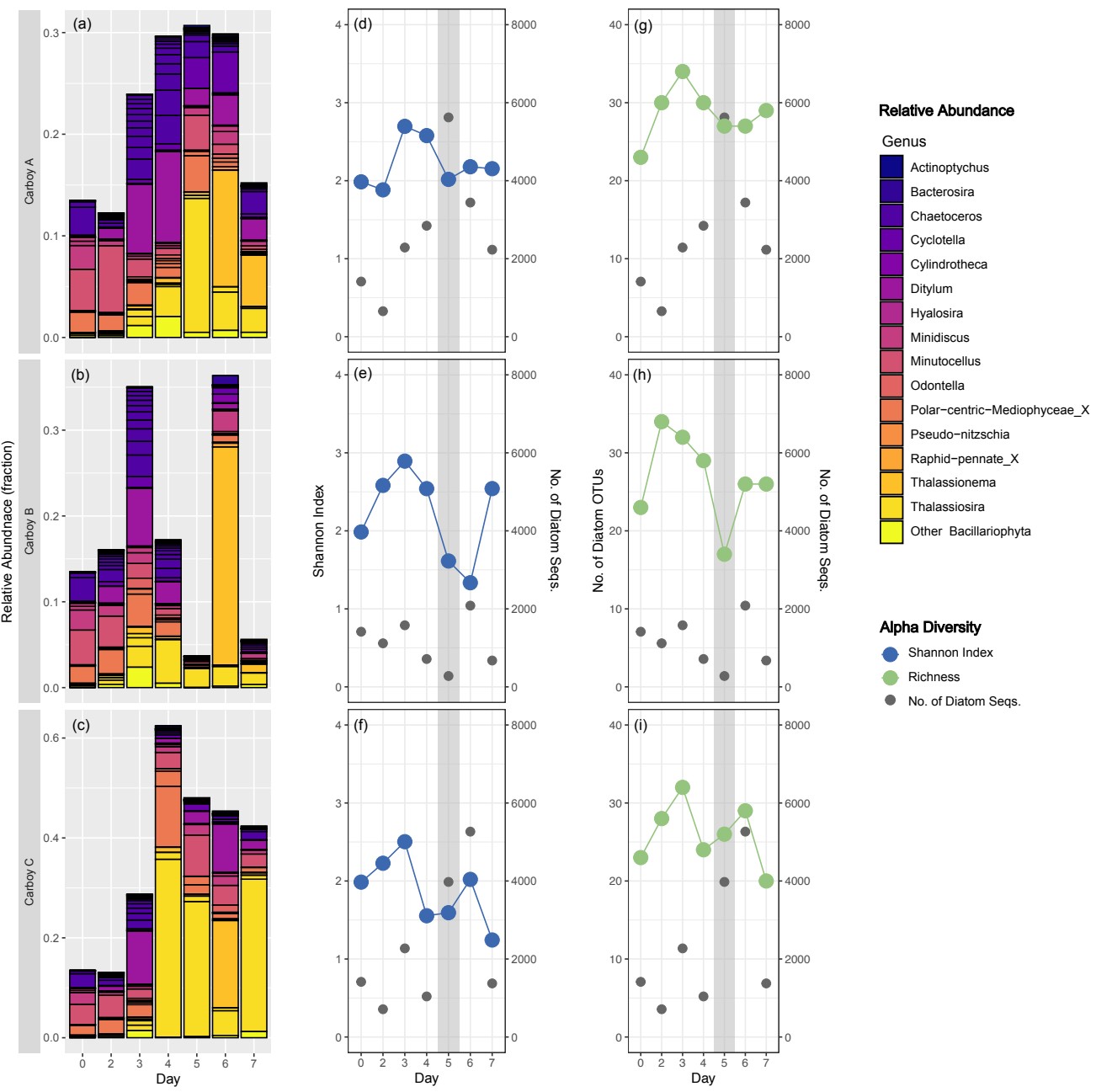

**Figure 5: Diatom community succession and alpha diversity.** (a–c) Relative abundance of diatom OTUs (fraction of the total non–metazoan eukaryote community) for each carboy, colored by genus with outlines around individual OTUs. "Day" corresponds to the number of days since inoculation, with day 0 for each carboy represented by the same merged inoculum sample. Y–axis range is variable. (d–i) Shannon alpha diversity (blue), number of diatom OTUs present (i.e. richness) (green), and the number of diatom sequences (grey) on the same "Day" scale. The grey shaded region indicates the peak bloom.

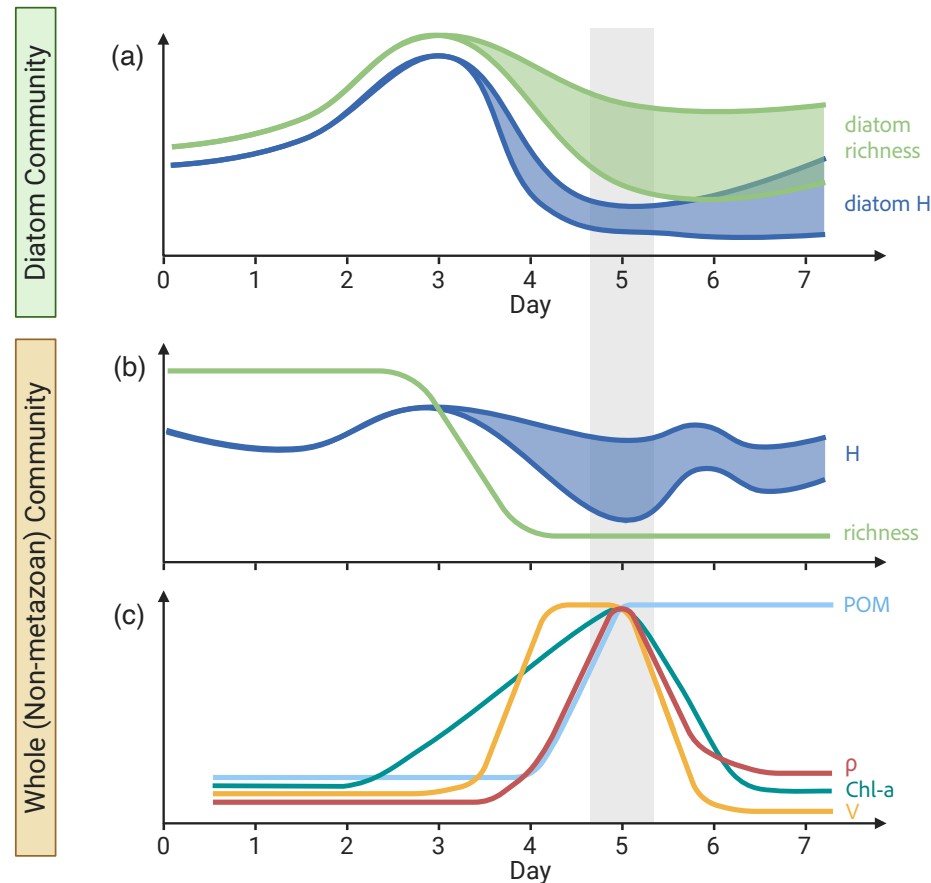

**595**

**Figure 6: Bloom summary.** Schematic diagram of the temporal trends in (a) diatom–specific 18S diversity, (b) whole non–metazoan eukaryotic community 18S diversity, and (c) biogeochemical measurements throughout the bloom experiments. Diversity is displayed as the number of OTUs (richness) and Shannon Index (H). Particulate organic matter (POM), chlorophyll a concentration (Chl-a), transport rates (ρ) and specific uptake rates (V) were chosen as the primary biogeochemical parameters. Trends are averaged across carboys and
**600** colored shaded regions represent deviations between carboys. Grey shaded region indicates the peak bloom. This figure was created in BioRender.

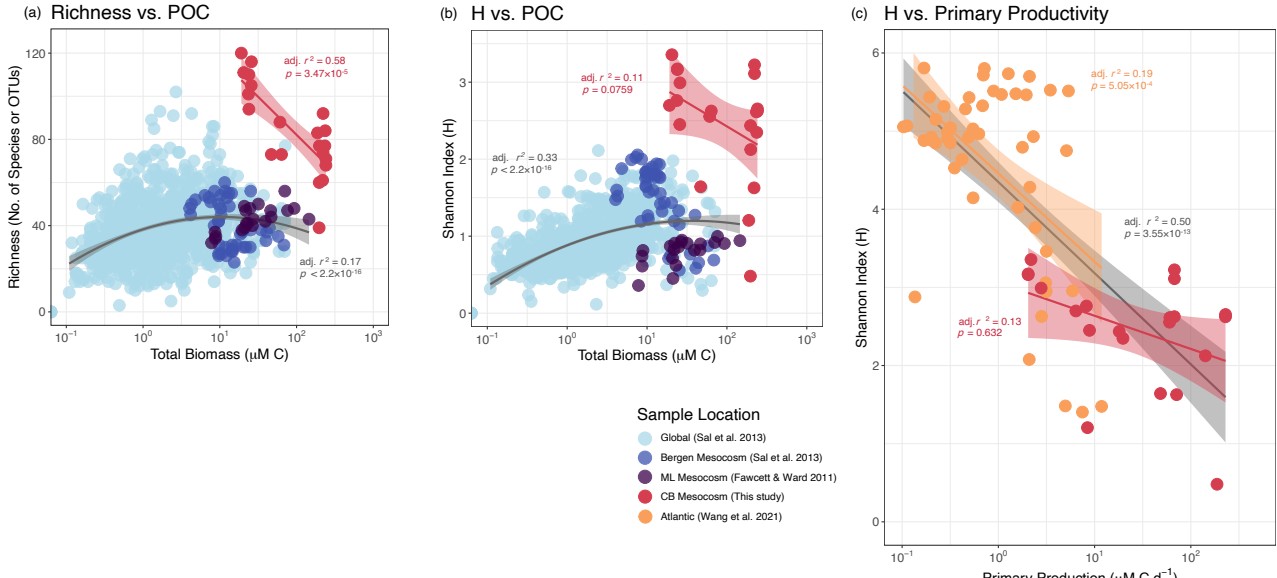

**Figure 7: Productivity–Diversity Relationship (PDR).** Several studies were compared using 3 different PDRs. (a,b) The global (Sal et al., 2013) and the Moss Landing (ML, Fawcett and Ward 2011) diversity data were obtained via microscopy and the Bergen mesocosm data was extracted from the global dataset. (c) The Atlantic (Wang et al., 2021) and Chesapeake Bay (CB, this study) diversity data were calculated from 18S–based OTUs. The Atlantic Net Community Production rates were converted to Primary Production rates according to Li & Cassar (2016, eq. 7). Quadratic regression curves, representing a 2D simplification of the Irigoien et al. (2004) 3D model, and linear regressions are plotted for select datasets. Shaded regions around linear regressions indicate the 95 % confidence interval and adjusted $r^2$ and model significance are listed in the respective color of a given dataset. The grey curves in (a,b) are fitted to all microscopy–based diversity data and the grey line in (c) is the linear regression for the combined Atlantic and Chesapeake Bay datasets.

## Data availability

The 18S data presented in the paper are available at NCBI SRA, BioProject ID: PRJNA1222857 (http://www.ncbi.nlm.nih.gov/bioproject/1222857).

The biogeochemical data presented in the paper are available at BCO-DMO, project number: 869541 (https://demo.bco-dmo.org/project/869541).

## Supplement

Detailed methodology and analyses, as well as supplemental tables and figures are available in the Supplementary Material.

## Author contributions (CRediT)

All authors were responsible for writing – review and editing (equal). Additional contributions:

JAL was responsible for conceptualization (supporting), data curation (lead), formal analysis (lead), investigation (lead), methodology (equal), visualization (lead), and writing – original draft (lead).

JHV was responsible for methodology/software (supporting) and supervision (supporting).

MAP was responsible for methodology/software (supporting) and writing – original draft (supporting).

LR was responsible for funding acquisition (supporting) and supervision (supporting).

BBW was responsible for conceptualization (lead), funding acquisition (lead), methodology (equal), project administration (lead), resources (lead), supervision (lead), and writing – original draft (supporting).

## Competing interests

The authors declare that they have no conflict of interest.

## Acknowledgements

We thank the captain and crew of the R/V Hugh Sharp for assistance during the Chesapeake Bay cruise in August 2021. Additional thanks to Ashley Maloney for her assistance in the setup of the microcosm experiment. Figure 6 was created in BioRender. Lee, J. (2025) https://BioRender.com/c17m525.

## Financial support

This research was supported by NSF grant OCE-2149606, awarded to B.B. Ward. Additional support to M.A. Poupon was
635 provided by NSF grant OCE-2023108, awarded to L. Resplandy.

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
