# Peer review of "Phytoplankton community succession and biogeochemistry in a bloom simulation experiment at an estuary-ocean interface"

_EGUsphere, 2025_

## Author Comment (AC1)

We thank the reviewers for their thoughtful and insightful comments on the manuscript. We have provided our responses in blue text and included some proposed changes to the wording of the main text/captions in *green italicized text* below.

**Reviewer 1 (Citation**: https://doi.org/10.5194/egusphere-2025-871-RC1**)**
**General Comments:**

This study conducted a microcosm experiment using a natural Chesapeake Bay phytoplankton community to evaluate bloom dynamics over time. The authors present a thorough and well-written manuscript which captures trends in diversity, particulates, and stoichiometry during their 8-day incubation. Their data clearly shows bloom initiation and termination, as well as a negative productivity-diversity relationship, and provides well thought out explanations for each of these observations. However, there is a strong focus on grazing as the cause of bloom termination, which seems to be driven by results from the COBALT simulation. Though there is some support for this conclusion, I would urge the authors to be more cautious as I don't believe there is sufficient data on grazing (rates, community etc.) to conclude that grazing was the main driver of bloom termination, though it may have played a role. Conversely, there seems to be a lot of data to support the impact of nutrient depletion on bloom dynamics (uptake rates, POC:Chl etc.), but this explanation was not fully explored. At elevated summer temperatures and in a closed system, nutrient depletion is likely to have contributed, and these factors should be considered in greater depth. Additionally, the authors could also discuss how a closed-system, like the one presented here, might compare to dynamics in the open ocean.

Overall, this study was compelling and well-constructed, incorporating diverse methodology, from DNA to an NPZ model, in order to evaluate phytoplankton bloom dynamics. In particular, this study provides valuable insight into how phytoplankton diversity changes throughout a bloom. Some additional discussion could make this study even more broadly applicable.

Thank you for your kind words and valuable input! We agree that the study would benefit from a deeper discussion of the role of nutrient limitation in our microcosm experiment. One analysis that we will add is the modeled nitrate limitation. The $N_{lim}$ parameter (eqn. S6) determines the amount of phytoplankton growth permitted by the available nutrients as a fraction of potential growth, where $N_{lim} = 1$ indicates no nitrate limitation and $N_{lim} = 0$ indicates complete limitation. Therefore $1 - N_{lim}$ represents the limitation and has been added to figure 3b as below.

[Figure]

We had initially anticipated that nutrient depletion would be the driver of the bloom's demise because the 210 μm inoculum pre-filter was meant to prevent large grazers. However, we came to suspect grazing as a factor due to the observed "gap" in N mass balance (prior to modeling), leading us to use the model to investigate that possibility further. We have detailed in our responses below other ways which we will be addressing nutrient depletion. We will also add a note on the impacts of a closed system experiment to section 4.6.

**Specific Comments:**

The methods are described well and are generally easy to follow.

Line 91: Were nutrients ever replenished in these microcosms or just spiked initially?

We will add clarification that the incubations did not receive any further nutrient additions.

*Line 91 → "Carboys were incubated for 8 days … without further nutrient additions in an on-deck…"*

Line 93: What was the exact temperature during the deck-board incubations?

The temperature data were presented in figure S7, but not previously discussed in the main text. We will add a sentence to the beginning of the results (line 212) stating the range of light and temperature measurements

*Line 93 → "Light (photosynthetically active radiation; PAR) and temperature displayed typical diel cycles during the incubations ranging 22.5-29ºC, with average daily temperature$_{max}$ = 28.0 ºC and average daily PAR$_{max}$ = 77.2Wm$^{-2}$ (median = 59.3 Wm$^{-2}$) (Fig. S7). Daytime PAR was lowest on day 3 and reached a maximum of 242.8 Wm$^{-2}$ at 14:00 on day 7."*

We will also clarify in the methods (line 93) that the continuous measurements were only taken between 18:00 on day 1 and 18:00 on day 7. Fig S7 will be updated to show this temporal range and the caption will be updated to clarify data averaging.

[Figure]

*Fig. S7 caption* → *"**Figure S7: Light and temperature.** Continuous light (yellow shaded region) and temperature (red line) measurements are plotted for days 1-7 of the microcosm experiment. Duplicate loggers were combined and measurements were averaged along hourly intervals (e.g. the value plotted for 12:00 is the average of all measurements taken between 11:30 and 12:30 from both data loggers). Note scale change above the y-axis break."*

Line 105: This section is unclear. As I understand, DNA analyses were conducted for all carboys and technical duplicates were done for carboy C only. However, this paragraph suggests that DNA was only analyzed from carboy C. Please reword so this can be clarified.

Thank you for bringing this to our attention. This will be clarified in the methodology.

*Line 105* → *"DNA samples were taken from each carboy concurrently with the 12:00 pigment and nutrient samples. Additional duplicates were collected on select days for carboy C and two additional DNA samples were collected from the surface water inoculum."*

Line 157: Is it reasonable to include a medium-sized copepod when these were filtered out from the microcosm (i.e. how big is the model grazer)? Why not a small zooplankton (e.g. small copepod or heterotrophic dinoflagellate)? How might this have impacted modeled grazing rates or grazing preferences?

The medium sized zooplankton (copepods) in the model represent the parameters of ~200–2,000 µm equivalent spherical diameter zooplankton (Stock et al., 2020). Given that the inoculum was pre-filtered with a 210 µm mesh, but the DNA analysis found Maxillopoda spp. and *Acartia tonsa* (which can grow much larger than 200 µm), we suspect that the zooplankton community started small and grew over the course of the bloom. Alternatively, the small zooplankton class have a higher/faster ingestion rate, and would likely have resulted in a shorter bloom with lower peak chlorophyll. Therefore, the medium size class was selected as an appropriate estimate of the dominant grazer activity.

While it might be more realistic to have used a range of phytoplankton and zooplankton sizes, the goal of this study was to test if a simplified NPZ model could replicate the observations and provide general insight on potential community dynamics.

Line 165: Did this assumption of phytoplankton biomass include an estimate of growth from day 1 to 2? What was the reasoning for not measuring chlorophyll on day 1?

The model day 1 phytoplankton biomass did not explicitly include growth between days 1 and 2. Instead, an average concentration of $N_{Phyto}$ = 5 µmol kg$^{-1}$ ± 20% was chosen to balance an initial phytoplankton population which was large enough to trigger a bloom, but small enough to roughly match day 2 chlorophyll concentrations.

Pigment samples were not collected on day one because we expected pigment concentrations to be below detection within the first 24 hours of incubation, so we did not take time zero or day 1 samples given our sample volume restrictions. As a result, we don't have individual carboy data for that day. However, a pseudo day 0 pigment analysis has been added to figure 1 (now showing the avg. pigment data across carboys). Two day 0 samples were produced by filtering ~1L each of a 10% dilution of inoculum in 0.3um filtered media immediately after inoculation and without receiving any additional nutrients. These details will be added to the methodology and clarified in the figure caption.

[Figure]

*Fig. 1 caption ➔ "**Figure 1: Biogeochemical data indicate a bloom.** (a) Particulate organic carbon (POC) and nitrogen (PON) concentrations for all carboys over the course of the bloom simulation. POC is in red/pink, PON is in blue, and line type denotes carboy. Error bars in (a)*

*represent the standard deviation of sample replicates. Average carboy concentrations are shown for (b) nutrients and (c) pigments (data for individual carboys are presented in Fig. S1). (b) Nutrient measurements during the bloom simulation are presented as dark blue lines for nitrate ([NO$_3^-$]), light blue lines for nitrite ([NO$_2^-$]), and purple lines for silicate ([SiO$_4^{4-}$]). (c) Pigment concentrations for chlorophyll a (Chl-a), chlorophyll c (Chl c1 + c2), and diagnostic pigments for diatoms (fucoxanthin), dinoflagellates (peridinin), and cyanobacteria (zeaxanthin). Error bars in (b, c) represent the standard deviation between carboys. Day 0 pigment concentration is from a separate 10% inoculation "dilution" (described in the methodology), rather than individual carboys. Time is shown as days since carboy inoculation where day 1 begins at 00:00 following inoculation. The grey shaded region indicates the peak bloom (~noon day 5)."*

Figure 1. This figure is nicely laid out to illustrate dynamics in the microcosm. It might be worth considering to use an average across all carboys, as in panel a, rather than just focusing on carboy C. Alternatively, figure S1 could be substituted here.

We agree with the reviewer that carboy averages are more appropriate for nutrient and pigment concentrations and figure 1 has been updated accordingly.

Line 240. The POC:Chl increases on day 6 are also consistent with phytoplankton becoming nutrient limited. See Jakobsen and Markager (2016), L&O or Arteaga et al. (2016), Glob. Biogeochem. Cycles. Generally, under replete conditions, phytoplankton cells tend to allocate greater resources to chlorophyll synthesis and growth, resulting in an inverse C:Chl – nutrient relationship. So, these trends could be a nutrient limitation signal as well as a grazing signal.

Thank you for your input and additional references. The relationship between nutrient availability and phytoplankton POC:Chl-a was a factor we considered and we do believe that it played a role in both the low ratios observed during the peak bloom and high ratios toward the end of the bloom. However, ratios >1000 are unlikely to be due solely to variable phytoplankton POC:Chl-a. This will be clarified in the discussion section (line ~362), but we provide additional detail below.

Figure 3e may have also caused confusion because it previously incorrectly displayed maximum POC:Chl-a < 250 even though ratios exceed 1000 (as in fig S3d-f). This has been fixed!

Model outputs from Arteaga et al. (2016, fig. 9) and Behrenfeld et al. (2005, fig. 2, 3) predict maximum phytoplankton POC:Chl-a of ~200 and ~300, respectively, and only in very low productivity regions and seasons, with nearly no available nutrients. Additionally, observation-based studies have found maximum values < 350 (Jakobsen and Markager, 2016, and the references therein), and diatom culture experiments (Laws and Bannister, 1980, from Behrenfeld et al. 2005 fig 3c) reported maximum POC:Chl-a < 500 in their highest nutrient stress and lowest growth rate conditions. Therefore, while phytoplankton-specific POC:Chl-a may have increased during the late-bloom, an increase non-phytoplankton biomass was also necessary to explain the observed ratios.

We will add to line 240 that phytoplankton-specific POC:Chl-a may also play a role in the high late-bloom ratios and clarify in the discussion (line ~363) why non-phytoplankton biomass must also be accumulating.

*Line 240 → "Late bloom ratios were much higher, reaching an average POC:Chl–a > 1000 on day 7, with maximum values of up to 1258.6 ± 247.2 in carboy B (Fig. S3d–f), suggesting the accumulation of non–phytoplankton biomass in the late bloom and a potential shift in phytoplankton POC:Chl–a."*

Line 276: How do you compare the low peridinin concentration with the high dinoflagellate relative abundance?

We were also initially surprised by the high relative abundance of dinoflagellates in the absence of peridinin. However, the primary dinoflagellates present in the DNA data were most closely related to *Karenia mikimotoi*, which do not use peridinin as their primary pigment. Based on additional accessory pigment analysis, we determined that the relative abundance of dinoflagellates was a result of high gene copy numbers (line 414-425). We go into more detail on the pigment analysis in the supporting material (Supporting analysis, lines S149-169).

Figure 4g – This figure is great. It shows changes over time, biological replication, and drivers all in one figure.

Thank you!

Line 352-355: This is a bit confusing. There only needs to be one limiting nutrient to cause a bloom decline. Here, the authors present evidence for a limiting nutrient on days 5 and 6, which is consistent with when the bloom crashes. It seems nutrients are being prematurely dismissed, but I would argue that they should be given greater focus and discussion in this manuscript.

We agree that the phrasing, especially "consistently limiting" was a bit confusing. We will update the text to note that nutrients may have been transiently limiting following the Liang et al. (2019) thresholds – i.e. met criteria for limitation in one sample, but not the following timepoint and not in multiple carboys at the same time, with the exception of the two timepoints already listed in the main text.

Additional discussion of nutrient's role in the bloom's demise will be added here, both in terms of the threshold-based limitation and the model estimations of nitrate limitation (added to fig 3).

*Line 352 → However, following the combined kinetics– and stoichiometry–based thresholds outlined in Liang et al. (2019), nutrients were only transiently limiting during the latter half of the bloom (Table S7). There were only two timepoints when nutrients were limited in more than one carboy: $SiO_4^{4-}$ on the evening of day 5 and $NO_3^-$ on the evening of day 6, and neither of these nutrients were limited in back–to–back samples. Additional analysis of the NPZ model revealed that nutrients likely became partially limiting following the bloom peak. Modeled nutrient limitation did not match the timing of threshold–based limitation, but did reach an average maximum of ~60% at the end of the blooming period (Fig. 3b). The combination of incomplete or transient nutrient limitation and consistently high POM during the late–bloom indicate that factors other than nutrient availability likely contributed to the bloom decline; the potential role of grazing is further investigated below.*

Line 363 – Again, POC:Chl can also be a sign of nutrient limitation. Please review the above references.

We agree that nutrient limitation likely led to increased phytoplankton POC:Chl-a during the late-bloom, but must conclude that an accumulation of non-phytoplankton biomass was also necessary to explain the observed POC:Chl-a > 1000. As noted above, we will add additional clarification for our reasoning.

*Line 362 → "In contrast to the low peak bloom POC:Chl–a, the extreme POC:Chl–a values observed at the end of the carboy incubations far exceeded both the average phytoplankton POC:Chl–a of ~40–90 observed in Chesapeake Bay (Cerco 2000 and the references therein) and previous observations of maximum phytoplankton POC:Chl–a <500 due to nutrient limitation (e.g.; Jakobsen and Markager, 2016; Laws and Bannister, 1980), with average day 7 values over 10–fold greater than expected for the region. Low available nutrients during the late–bloom may have contributed to an increase in POC:Chl–a, however high total community POC:Chl–a is primarily influenced by the ratio of phytoplankton POC to zooplankton and detrital POC (Banse 1977)."*

Line 365: Could these metazoan sequences result from copepod detritus?

It is possible that a portion of the metazoan sequences were from detritus or external DNA. However, if metazoan DNA was primarily present detritus, we would expect the relative abundance of all metazoan OTUs to be highest at the beginning and decrease over time, as the DNA is degraded and the POM of living organisms increased. Instead, our 3 arthropod OTUs have distinct relative abundance patterns – peaking at different points in the bloom.

In the figure below, sp1 is a Maxillopoda spp., sp3 is *Acartia tonsa*, and sp134 is a combination of "other" arthropods.

[Figure]

Lines 470-477: It may also be worth noting that this incubation was a closed system design and thus likely is unable to capture all the diversity patterns that exist in an open system. A closed

system prevents both immigration/emigration and nutrient replenishment which could have impacts on diversity metrics.

We agree that micro- and mesocosm experiment alpha diversity is bounded by the organisms present in the initial inoculation. This will be noted in the text.

*Line 472 → "The issue of regionality may occur because a range of several orders of magnitude ... because taxa appear more cosmopolitan at small scales (Smith 2007). The latter effect may be heightened in micro- and mesocosm experiments where diversity is bounded on both ends by the inoculum community. It is also true for many generalist microbes regardless of study scale and may also explain why ..."*

Line 497: While grazing may have contributed significantly to the observed trends, I still think its important not to discount the role that nutrients may have had on bloom termination. This is briefly stated on lines 509-511, but could be expanded on throughout.

Agreed; the impact of nutrient limitation will also be noted in the conclusion.

**Technical Corrections:**

Line 36: Maybe "silica cell walls" instead of "silica shells?"

We will correct "shells" to "cell walls."

Line 336-337: This could be moved to the results.

Thank you for pointing this out. We will move the significance values (Kruskal-Wallis p-values) to the results section (line 252).

References: Sal et al 2013 (Figure 7) is missing from the references.

Thank you for catching this. The citation will be added to the references!

Line 470: It's difficult to distinguish the unimodal relationship in the global dataset. Could the points be made transparent (in R, use 'alpha'), to help visualize the density of points?

We will clarify in the main text that the canonical unimodal PDR is defined not only by well delineated curves, but also by clusters of points bounded by a unimodal curve (Smith, 2007). As noted in Skácelová & Lepš (2014), "biomass can be important in regulating the upper limit of diversity, whereas at all the biomass values, extremely species poor communities are found." We will also note that the well delineated unimodal curve in Irigoien et. al. (2004) showed that phytoplankton- or zooplankton-specific diversity was a function of both phytoplankton and zooplankton biomass as separate parameters. Collapsing Irigoien et al.'s 3D model (2004, fig2c shown below) into two dimensions broadens the range of expected diversity at a given biomass.

[Figure]

*Line 470 → "While a unimodal PDR, either expressed as a unimodal curve or a cloud of points whose upper bound is defined by a unimodal curve (Skácelová and Lepš, 2014; Smith, 2007), may be observed in regional studies, it may not apply to local diversity patterns, which can appear monotonic (Rosenzweig, 1992). This can be seen when comparing the combined global and local data as a whole to individual localized experiments (Fig. 7a,b). Excluding the much higher 18S-derived diversity of this study, the upper bounds of all microscopy-derived data clearly displayed unimodal patterns and the data fit flatter, though still significant, unimodal curves. Additionally, Irigoien et. al. (2004) showed that phytoplankton- or zooplankton-specific diversity was a function of both phytoplankton and zooplankton biomass as separate parameters, making the unimodal PDR curve of combined factors less distinctly defined. Contrastingly, the individual localized micro- and mesocosm studies displayed both positive and negative monotonic PDRs. The issue of regionality…all commonly reported in aquatic microbial PDR studies (Graham and Duda, 2011; Skácelová and Lepš, 2014; Smith, 2007). It is important to note that the potential diversity at any given time in incubation studies is bounded by the organisms present in the inoculum. However, aside from unimodal…natural aquatic microbial communities (Smith, 2007)."*

We can also add a quadratic regression curve fitted to all microscopy-derived diversity data, to figure 7a,b by modifying the equation outlined in Irigoien et al. (2004). We will note in the caption and the main text which data is included in the curve, and in the caption that it is representative of 3D model simplified to 2 dimensions. The colors and alpha in figure 7 will be changed to make visualization clearer and the colors figure S6 will be changed for consistency.

[revised manuscript text omitted]

---

## Author Comment (AC2)

We thank the reviewers for their thoughtful and insightful comments on the manuscript. We have provided our responses in blue text and included some proposed changes to the wording of the main text/captions in *green italicized text* below.

**Reviewer 2 (Citation**: https://doi.org/10.5194/egusphere-2025-871-RC2)
**General Comments:**
This study presents the results from a microcosm experiment conducted to follow a natural phytoplankton bloom in Chesapeake Bay, using various methodological approaches (including 18S rRNA gene analysis and an NPZ model). The introduction in well-written, with a clear focus and a plausible research gap and objective. The methods are very thoroughly described and good to follow. All figures are done very nicely and informatively. I added some comments throughout the methods and introduction to consider. For the discussion, I agree with reviewer #1 that the depletion of nutrients for terminating the bloom should receive a bit more thought (as scratched upon in lines 351-355) and I added some additional comments.
Thank you for your thoughtful feedback! We agree with both reviewers that the study would benefit from a deeper discussion of the role of nutrient limitation in our microcosm experiment. We have detailed in our responses how we will be addressing nutrient depletion.

**METHODS**
Line 87-89: How were the added concentrations of nitrogen, silica and phosphorus chosen? The reasoning for their supply ratio is clear with the follow-up sentence, but I am wondering about their concentration.
40 μM $NO_3^-$ was chosen to promote a diatom bloom, as well as to represent the nutrient loading to which Chesapeake Bay is exposed. Silica and phosphorus concentrations were chosen to provide appropriate nutrient ratios. Lower nutrient concentrations can promote cyanobacterial or dinoflagellate blooms over a diatom bloom (Adolf et al., 2006; Conley and Malone, 1992; Huang et al., 2020). While 40 μM $NO_3^-$ is a little high compared to historical ambient nutrient concentrations at the mouth of the bay, it is well within the range of observed nutrients and blooming concentrations observed throughout the main stem of Chesapeake Bay (Harding et al., 2019, fig. 4g-i; Malone et al., 1996, fig. 6).
Line 100 and line 165: Why were pigment and chlorophyll measurements only started/shown on day 2? In the discussion, it says (in lines 396-398): "The variability in the timing of the bloom peak may be due to minor differences in the starting community that each carboy received, as seen in the dissimilarity present between replicate inoculum samples despite being filtered from the same stock of water." To get an own impression of this, it would be informative to see day 0 data for all carboys.
Reviewer #1 had the same query and a more detailed explanation is presented in our response to their comments. Pigment samples were not collected from individual carboys until day 2 because we expected pigment concentrations to be below detection early in the bloom and did not have enough water in the experimental incubations to allow filtration of a larger volume to increase

assay sensitivity. Instead, we have added pseudo day 0 pigment data to figure 1 (as shown below).

[Figure]

Regarding lines 396-398, this is referring the beta diversity analysis of 18S-based community composition seen in figure 4g. We thank the reviewer for pointing out that this needs clarification and will add a reference to Fig. 4g here.

Lines 104-105: The DNA samples at 12:00 were collected for all carboys? Maybe to make it clearer add that it was all carboys in that case. How did you select the days on which you additionally sampled carboy C i.e., what additional information do you gain from those additional days?

Thank you to both reviewers for pointing out the confusing wording here. We will add clarification that duplicate inoculation samples were collected for day 0, each carboy was sampled daily at ~noon, and that carboy C was additionally sampled for duplicates on certain days. We have provided proposed changes in our response to reviewer #1.

Due to constraints in sequencing capacity, we selected three relatively high biomass samples around the expected bloom transition and peak, where we expected the greatest variability. This allowed us to investigate how the magnitude of carboy-to-carboy community variability compared to potentially patchy sampling within a carboy.

Line 165: What is the assumption of a 1:100 biomass ratio of zooplankton to phytoplankton based on? Is there any literature on this that can be referenced here?

The 1:100 biomass ratio was chosen based on the assumption that the concentration of zooplankton would be comparatively low at the start of the bloom due to the 210 μm inoculum pre-filter, which would have removed large zooplankton while allowing most phytoplankton to pass through. We note that 1:100 is just the average ratio used for initialization, as the $N_{NO3}$, $N_{Phyto}$, and $N_{Zoo}$ each varied ± 20% across model iterations.

Line 192: Which alpha diversity metric was calculated and why?

Both OTU richness and Shannon index (H) were used as alpha diversity metrics to compare our data to previous PDR studies, as the observed PDR can vary depending on if/how a diversity measure accounts for evenness. We did not initially think it was necessary to specify this in the methodology, as we often refer to general trends in alpha diversity in the results and discussion. We can add clarification in the methodology for the alpha diversity metrics used.

*e.g. Line 192 → "For relative abundance and alpha diversity analyses (OTU richness and Shannon index), carboy C duplicates…"*

**RESULTS**

Fig. 1: Why are nutrients and pigments only shown for carboy 3? I suggest including the other carboys as well. As written the text and as also seen in Fig S1, the carboys indeed behave quite similarly. But with only showing one carboy, it always seems a bit suspicious to me at first.

Thank you to both reviewers for pointing this out. Figure 1 has been updated to show carboy averages for nutrients and pigments (shown above).

Lines 221-222: I think, it is not 100% correct to say that phosphate followed a similar pattern as the other nutrients. In the next half sentence, it is already stated that different from the other nutrients, phosphate gradually decreased while the other nutrients stayed rather constant until day 4 and then rapidly decreased. Consider rephrasing the start of the sentence.

The description of phosphate concentration patterns will be reworded.

*Line 223 → "Phosphate ($PO_4^{3-}$) concentrations decreased more consistently between days 1 and 5 (Fig. S2) from an initial average concentration of 4.0 ± 0.4 μM to 0.6 ± 0.1 μM by 18:00 day 5."*

Fig. 1: Chlorophyll c which is present in the Figure 1 is not mentioned in the text, although all other pigments are mentioned. Please clarify.

Variable ratios of Chl-c to other pigments throughout the bloom can be indicative of a community shift (e.g. Chl-a to Chl-c ratio changes during the late bloom). This will be noted in the text.

*Line 236 → "Pigment accumulation trends along with the concurrent consumption of $SiO_4^{4-}$ indicate that a diatom bloom occurred, and the late–bloom decrease in Chl–a:chlorophyll c may indicate a shift in phytoplankton community following the bloom peak (Dursun et al., 2021). Additionally, the decoupling between Chl–a and POM concentrations in the late bloom resulted in a large range of POC:Chl–a ratios."*

Lines 265-269: This part can be moved to the methods.

We had initially included this short paragraph to give a brief review of how OTUs were defined in our study and provide an explanation for why metazoan sequences were removed from the results presented in this section. This information is also provided with greater detail in the methodology. If the reviewers feel that this information does not need to be repeated in the results, we can remove it.

Fig. 4: Just leaving this comment here with knowing this is hard to change. While reading the paragraph from lines 274-281, I realized I am not able to tell the classes in Fig. 4 apart myself due to very similar colors. As I said, I am just leaving this here as a note.

We appreciate this note and the understanding from our reviewers. While we found this color palette to have the best color variability while remaining accessible to those with color vision deficiencies, we are open to suggestions for different color palettes.

Fig. 4: The color of the inoculum shapes is hardly visible. I know the color in this case is redundant with the shape, but maybe changing the shape to a diamond would help here.

[Figure]

Thank you for pointing this out. The symbol for inoculum samples will be updated as shown above so that the colors are more visible.

**DISCUSSION**

Line 347-348; line 354-355: These two sentences read a bit like they open a line of reasoning but close it again without giving it enough credit. While the first one says that a depletion of nutrients on day 5 led to the bloom's demise, the last sentence says that factors other than nutrient limitation need to be considered for the bloom's termination. This leaves the question open why the nutrient limitation that is mentioned in the first part is not further discussed.

We agree with both reviewers that the role of nutrient depletion in the bloom's demise should be discussed more deeply. We will add discussion (proposed changes provided above in our response to reviewer #1) of transient nutrient limitation based on the kinetics- and stoichiometry-based thresholding, as well as partial nutrient limitation estimated by the NPZ model.

Line 433: When H was already lower than other studies, but not as low as expected during a bloom, were the other studies that are referenced here not during a bloom? Whether they measure H during a bloom or not already makes quite a difference, as also mentioned in the discussion.

The Wang et al. (2024) and Cram et al. (2024) studies did not specifically target blooms, but were chosen for comparison because they reported on 18S-based analyses of the whole eukaryotic community in the main stem of the bay. Cram et al. (2024) sampled throughout Chesapeake Bay during the summer, while Wang et al. (2024) sampled along both spatial and seasonal gradients. While Wang et al. (2024) included samples which may have been collected during seasonal blooms, they found no statistical difference in H across seasons (Wang et al., fig. 3) and rarely observed H < 3 (Wang et al., fig. 4).

Line 476-477: This statement needs some references, I think, even though some are mentioned before in the text.

We will add another citation for Smith 2007 and clarify that this is for aquatic systems.

**SMALL CORRECTION:**

Line 536: "(c) particulate organic carbon (POC) and …" instead of "particulate organic (c) carbon (POC)" in description of Fig. 3.

Thank you! We will update the wording.

*Fig. 3 caption → "**Figure 3:** ... and (c) particulate organic carbon (POC) and (d) particulate organic nitrogen (PON) concentrations. ..."*

**References**

Adolf, J. E., Yeager, C. L., Miller, W. D., Mallonee, M. E., and Harding, L. W.: Environmental forcing of phytoplankton floral composition, biomass, and primary productivity in Chesapeake Bay, USA, Estuar. Coast. Shelf Sci., 67, 108–122, https://doi.org/10.1016/j.ecss.2005.11.030, 2006.

Conley, D. and Malone, T.: Annual cycle of dissolved silicate in Chesapeake bay: implications for the production and fate of phytoplankton biomass, Mar. Ecol. Prog. Ser., 81, 121–128, https://doi.org/10.3354/meps081121, 1992.

Cram, J. A., Hollins, A., McCarty, A. J., Martinez, G., Cui, M., Gomes, M. L., and Fuchsman, C. A.: Microbial diversity and abundance vary along salinity, oxygen, and particle size gradients in the Chesapeake Bay, Environ. Microbiol., 26, e16557, https://doi.org/10.1111/1462-2920.16557, 2024.

Dursun, F., Tas, S., and Ediger, D.: Assessment of phytoplankton group composition in the Golden Horn Estuary (Sea of Marmara, Turkey) determined with pigments measured by HPLC-CHEMTAX analyses and microscopy, J. Mar. Biol. Assoc. U. K., 101, 649–665, https://doi.org/10.1017/S0025315421000631, 2021.

Harding, L. W., Mallonee, M. E., Perry, E. S., Miller, W. D., Adolf, J. E., Gallegos, C. L., and Paerl, H. W.: Long-term trends, current status, and transitions of water quality in Chesapeake Bay, Sci. Rep., 9, 6709, https://doi.org/10.1038/s41598-019-43036-6, 2019.

Huang, K., Feng, Q., Zhang, Y., Ou, L., Cen, J., Lu, S., and Qi, Y.: Comparative uptake and assimilation of nitrate, ammonium, and urea by dinoflagellate Karenia mikimotoi and diatom

Skeletonema costatum s.l. in the coastal waters of the East China Sea, Mar. Pollut. Bull., 155, 111200, https://doi.org/10.1016/j.marpolbul.2020.111200, 2020.

Malone, T. C., Conley, D. J., Fisher, T. R., Glibert, P. M., Harding, L. W., and Sellner, K. G.: Scales of Nutrient-Limited Phytoplankton Productivity in Chesapeake Bay, Estuaries, 19, 371, https://doi.org/10.2307/1352457, 1996.

Wang, H., Liu, F., Wang, M., Bettarel, Y., Eissler, Y., Chen, F., and Kan, J.: Planktonic eukaryotes in the Chesapeake Bay: contrasting responses of abundant and rare taxa to estuarine gradients, Microbiol. Spectr., 12, e04048-23, https://doi.org/10.1128/spectrum.04048-23, 2024.

---

## Author Response (AR1)

We thank the reviewers for their thoughtful and insightful comments on the manuscript. We have synthesized our point-by-point responses in blue text and provided the relevant changes to the manuscript in *green italicized text* below. Line numbers referenced in our responses correspond to the track-changes version of the updated manuscript.

**Reviewer 1 (Citation**: https://doi.org/10.5194/egusphere-2025-871-RC1)
**Reviewer 2 (Citation**: https://doi.org/10.5194/egusphere-2025-871-RC2)

**R1 & R2 General Comments:**

We thank both reviewers for their kind words and valuable input! We agree that the study would benefit from a deeper discussion of the role of nutrient limitation in our microcosm experiment. We had initially anticipated that nutrient depletion would be the driver of the bloom's demise because the 210 μm inoculum pre-filter was meant to prevent large grazers. However, we came to suspect grazing as a factor due to the observed "gap" in N mass balance (prior to modeling), leading us to use the NPZ model to investigate that possibility further. We have expanded on role of nutrient limitation throughout the updated manuscript, but especially in discussion section 4.1, and have detailed in our responses below how we have addressed nutrient depletion.

Primarily, we have added an analysis of the nitrate limitation parameter in the NPZ model ($N_{lim}$, eq. S6), which determines the amount of phytoplankton growth permitted by the available nutrients as a fraction of potential growth, where $N_{lim} = 1$ indicates no nitrate limitation and $N_{lim} = 0$ indicates complete limitation. Therefore $1 - N_{lim}$ represents the limitation and has been added to figure 3b (see our response to R2 Line 347-348; line 354-355). We have also noted the role of nutrient limitation in the phytoplankton POC:Chl-a and explained why an accumulation of non-phytoplankton biomass is also necessary to explain observations in discussion section 4.2 (see out response to R1 Line 363).

We have also noted the impacts of a closed system experiment to sections 4.3 and 4.6 (detailed in our response to R1 Lines 470-477 below).

**Specific Comments:**

- R1 Line 36: Maybe "silica cell walls" instead of "silica shells?"
  We have changed "shells" to "cell walls."
  *Updated line 37 → "…but also because they have silica cell walls which may reduce grazing pressure…"*

- R1: The methods are described well and are generally easy to follow.
  Thank you!

- R2 Line 87-89: How were the added concentrations of nitrogen, silica and phosphorus chosen? The reasoning for their supply ratio is clear with the follow-up sentence, but I am wondering about their concentration.
  40 μM $NO_3^-$ was chosen to promote a diatom bloom, as well as to represent the nutrient loading to which Chesapeake Bay is exposed. Silica and phosphorus concentrations were

chosen to provide appropriate nutrient ratios. Lower nutrient concentrations can promote cyanobacterial or dinoflagellate blooms over a diatom bloom (Adolf et al., 2006; Conley and Malone, 1992; Huang et al., 2020). While 40 µM $NO_3^-$ is a little high compared to historical ambient nutrient concentrations at the mouth of the bay, it is well within the range of observed nutrients and blooming concentrations observed throughout the main stem of Chesapeake Bay (Harding et al., 2019, fig. 4g-i; Malone et al., 1996, fig. 6). This has been clarified in the main text.

*Updated line 91 → "40 µM $NO_3^-$ was chosen to mimic historical observations of nutrient loading in Chesapeake Bay (Harding et al., 2019; Malone et al., 1996) and promote a diatom bloom."*

- R1 Line 91: Were nutrients ever replenished in these microcosms or just spiked initially? We have clarified in the text that the incubations did not receive any further nutrient additions.

  *Updated line 94 → "Carboys were incubated for 8 days without further nutrient additions in an on-deck…"*

- R1 Line 93: What was the exact temperature during the deck-board incubations? We have added a sentence to the beginning of the results stating the range of light and temperature measurements.

  *Updated line 234 → "Light (photosynthetically active radiation, PAR) and temperature displayed typical diel cycles during the incubations, ranging 22.5–29°C with average daily $temperature_{max}$ = 28.0°C and average daily $PAR_{max}$ = 77.2$Wm^{-2}$ (median = 59.3 $Wm^{-2}$) (Fig. S1). Daytime PAR was lowest on day 3 and reached a maximum of 242.8 $Wm^{-2}$ at 14:00 on day 7."*

  We have also clarified in the methods that the continuous measurements were taken between 18:00 on day 1 and 18:00 on day 7. Fig. S1 (previously Fig. S7) has been updated to show this temporal range and the caption has been updated to clarify data averaging.

  *Updated line 96 → "Continuous light and temperature measurements were recorded between 18:00 on day 1 and 18:00 on day 7 using two Onset HOBO Pendant Temperature/Light data loggers suspended ~10 cm below the surface of the on–deck water bath."*

[Figure]

*Updated Fig. S1 caption → **"Figure S1: Photosynthetically active radiation (PAR) and temperature.** Continuous light (PAR, yellow shaded region) and temperature (red line) measurements are plotted for days 1–7 of the mesocosm experiment. Duplicate loggers were combined and measurements were averaged along hourly intervals (e.g. the value displayed for 12:00 is the average of all measurements taken between 11:30 and 12:30 from both data loggers). Note scale change above the y–axis break."*

- R2 Line 100 and line 165: Why were pigment and chlorophyll measurements only started/shown on day 2? In the discussion, it says (in lines 396-398): "The variability in the timing of the bloom peak may be due to minor differences in the starting community that each carboy received, as seen in the dissimilarity present between replicate inoculum samples despite being filtered from the same stock of water." To get an own impression of this, it would be informative to see day 0 data for all carboys.

  Pigment samples were not collected on day one because we expected pigment concentrations to be below detection within the first 24 hours of incubation, so we did not take time zero or day 1 samples given our sample volume restrictions. As a result, we don't have individual carboy data for that day. However, a pseudo day 0 pigment analysis has been added to figure 1 (now showing the avg. pigment data across carboys). Two day 0 samples were produced by filtering ~1L each of a 10% dilution of inoculum in 0.3μm filtered medium immediately after inoculation and without receiving any additional nutrients. These details have been added to the methodology and clarified in the Fig. 1 caption.

  *Updated line 108 → "Pigment samples were not collected from carboys prior to day 2 due to low biomass concentrations and sample volume restrictions. Instead, two pseudo day 0 samples were produced by filtering ~1 L each of a 10 % inoculum dilution in 0.3 μm filtered medium immediately after inoculation and without receiving any additional nutrients."*

[Figure]

*Updated Fig. 1 caption → "**Figure 1:** ... Error bars in (a) represent the standard deviation of sample replicates. Average carboy concentrations are shown for (b) nutrients and (c) pigments (data for individual carboys are presented in Fig. S2) ... Error bars in (b, c) represent the standard deviation between carboys. Day 0 pigment concentrations are from a separate 10 % inoculation "dilution" (described in the methodology), rather than individual carboys. Time is shown as days since carboy inoculation where day 1 begins at 00:00 following inoculation. The grey shaded region indicates the peak bloom (~noon day 5)."*

Regarding lines 396-398, this is referring the beta diversity analysis of 18S-based community composition seen in figure 4g. Thank you for pointing out that this needed clarification. We added a reference to figure 4g on line 463.

*Updated line 463-464 → "...as seen in the dissimilarity present between replicate inoculum samples despite being filtered from the same stock of water (**Fig. 4g**)."*

- R2 Lines 104-105: The DNA samples at 12:00 were collected for all carboys? Maybe to make it clearer add that it was all carboys in that case. How did you select the days on which you additionally sampled carboy C i.e., what additional information do you gain from those additional days?

Thank you to both reviewers for pointing out the confusing wording here. We have added

clarification that duplicate inoculation samples were collected for day 0, each carboy was sampled daily at ~noon, and that carboy C was additionally sampled for duplicates on certain days.

Due to constraints in sequencing capacity, we selected three relatively high biomass samples around the expected bloom transition and peak, where we expected the greatest variability. This allowed us to investigate how the magnitude of carboy-to-carboy community variability compared to potentially patchy sampling within a carboy (i.e. Fig. 4g).

*Updated line 113 → "DNA samples were taken from each carboy concurrently with the 12:00 pigment and nutrient samples. Three samples were collected in duplicate for carboy C (on days surrounding the expected bloom transition and peak) and two DNA samples were collected from the surface water inoculum."*

- R1 Line 105: This section is unclear. As I understand, DNA analyses were conducted for all carboys and technical duplicates were done for carboy C only. However, this paragraph suggests that DNA was only analyzed from carboy C. Please reword so this can be clarified.

  Thank you for bringing this to our attention. We have clarified the wording in the methodology (see our response to R2 Lines 104-105 above)

- R1 Line 157: Is it reasonable to include a medium-sized copepod when these were filtered out from the microcosm (i.e. how big is the model grazer)? Why not a small zooplankton (e.g. small copepod or heterotrophic dinoflagellate)? How might this have impacted modeled grazing rates or grazing preferences?

  The medium sized zooplankton (copepods) in the model represent the parameters of ~200–2,000 μm equivalent spherical diameter zooplankton (Stock et al., 2020). Given that the inoculum was pre-filtered with a 210 μm mesh, but the DNA analysis found Maxillopoda spp. and *Acartia tonsa* (which can grow much larger than 200 μm), we suspect that the zooplankton community started small and grew over the course of the bloom. Alternatively, the small zooplankton class have a higher/faster ingestion rate, and would likely have resulted in a shorter bloom with lower peak chlorophyll. Therefore, the medium size class was selected as an appropriate estimate of the dominant grazer activity.

  While it might be more realistic to have used a range of phytoplankton and zooplankton sizes, the goal of this study was to test if a simplified NPZ model could replicate the observations and provide general insight on potential community dynamics.

  The supplemental methods have been updated to clarify the size range of modeled zooplankton.

  *Updated line S54 → "The zooplankton parameters are tuned to represent a medium–sized copepod population (~200–2,000 μm equivalent spherical diameter zooplankton)."*

- R1 Line 165: Did this assumption of phytoplankton biomass include an estimate of growth from day 1 to 2? What was the reasoning for not measuring chlorophyll on day 1?

Reviewer #2 had the same query about pigment concentrations before day 2 and a more detailed explanation is presented in our response to their comments (see R2 Line 100 and line 165 above). Pigment samples were not collected from individual carboys until day 2 because we expected pigment concentrations to be below detection early in the bloom and did not have enough water in the experimental incubations to allow filtration of a larger volume to increase assay sensitivity. Instead, we have added pseudo day 0 pigment data to figure 1 (shown above).

The model day 1 phytoplankton biomass did not explicitly include growth between days 1 and 2. Instead, an average concentration of $N_{Phyto}$ = 5 μmol kg$^{-1}$ ± 20% was chosen to balance an initial phytoplankton population which was large enough to trigger a bloom, but small enough to roughly match day 2 chlorophyll concentrations. This has been clarified in the methodology.

*Updated line 182 → "... an estimation of day 1 $N_{Phyto}$ (5 μmol kg$^{-1}$) from observed day 2 Chl–a concentrations... The average starting concentration of $N_{Phyto}$ was chosen to balance an initial phytoplankton population which was large enough to trigger a bloom, but small enough to roughly match day 2 chlorophyll concentrations."*

- R1 Figure 1. This figure is nicely laid out to illustrate dynamics in the microcosm. It might be worth considering to use an average across all carboys, as in panel a, rather than just focusing on carboy C. Alternatively, figure S1 could be substituted here.

  We agree that carboy averages are more appropriate for nutrient and pigment concentrations and Fig. 1 has been updated accordingly (see our response to R2 Line 100 and line 165 above).

- R2 Fig. 1: Why are nutrients and pigments only shown for carboy 3? I suggest including the other carboys as well. As written the text and as also seen in Fig S1, the carboys indeed behave quite similarly. But with only showing one carboy, it always seems a bit suspicious to me at first.

  Thank you to both reviewers for pointing this out. Fig. 1 has been updated to show carboy averages for nutrients and pigments (see our response to R2 Line 100 and line 165 above).

- R2 Fig. 1: Chlorophyll c which is present in the Figure 1 is not mentioned in the text, although all other pigments are mentioned. Please clarify.

  Variable ratios of Chl-c to other pigments throughout the bloom can be indicative of a community shift (e.g. Chl-a to Chl-c ratio changes during the late bloom). This has been added to the main text.

  *Updated line 262 → "... the late–bloom decrease in Chl–a:chlorophyll c may indicate a shift in phytoplankton community following the bloom peak (Dursun et al., 2021)."*

- R2 Line 165: What is the assumption of a 1:100 biomass ratio of zooplankton to phytoplankton based on? Is there any literature on this that can be referenced here?

  The 1:100 biomass ratio was chosen based on the assumption that the concentration of zooplankton would be comparatively low at the start of the bloom due to the 210 μm

inoculum pre-filter, which would have removed large zooplankton while allowing most phytoplankton to pass through. We note that 1:100 is just the average ratio used for initialization, as the $N_{NO3}$, $N_{Phyto}$, and $N_{Zoo}$ were individually varied ± 20% across model iterations.

- R2 Line 192: Which alpha diversity metric was calculated and why?
Both OTU richness and Shannon index were used as alpha diversity metrics to compare our data to previous PDR studies, as the observed PDR can vary depending on if/how a diversity measure accounts for evenness. We have added clarification in the methodology for the alpha diversity metrics used.
*Updated line 213 → "For relative abundance and alpha diversity analyses (OTU richness and Shannon index)…"*

- R2 Lines 221-222: I think, it is not 100% correct to say that phosphate followed a similar pattern as the other nutrients. In the next half sentence, it is already stated that different from the other nutrients, phosphate gradually decreased while the other nutrients stayed rather constant until day 4 and then rapidly decreased. Consider rephrasing the start of the sentence.
The description of phosphate concentration patterns has been reworded.
*Updated line 248 → "Phosphate ($PO_4^{3-}$) concentrations decreased more consistently between days 1 and 5 from an initial average of 4.0 ± 0.4 µM to 0.6 ± 0.1 µM by 18:00 day 5 (**Fig. S3**)."*

- R1 Line 240. The POC:Chl increases on day 6 are also consistent with phytoplankton becoming nutrient limited. See Jakobsen and Markager (2016), L&O or Arteaga et al. (2016), Glob. Biogeochem. Cycles. Generally, under replete conditions, phytoplankton cells tend to allocate greater resources to chlorophyll synthesis and growth, resulting in an inverse C:Chl – nutrient relationship. So, these trends could be a nutrient limitation signal as well as a grazing signal.
Thank you for your input and additional references. The relationship between nutrient availability and phytoplankton POC:Chl-a was a factor we considered and we do believe that it played a role in both the low ratios observed during the peak bloom and high ratios toward the end of the bloom. However, ratios >1000 are unlikely to be due solely to variable phytoplankton POC:Chl-a. This has been noted briefly in line 276 and discussed in greater detail in lines 382-386 (424-428 of the track-changes file).
*Updated line 276 → "…the accumulation of non–phytoplankton biomass in the late bloom and a potential shift in phytoplankton POC:Chl–a"*
*Updated line 423 → "… far exceeded both the average phytoplankton POC:Chl–a of ~40–90 observed in Chesapeake Bay … and previous observations of maximum phytoplankton POC:Chl–a < 500 (e.g.; Laws and Bannister, 1980; Jakobsen and Markager, 2016) … Low available nutrients during the late–bloom likely contributed to an increase in phytoplankton POC:Chl–a (e.g.; Behrenfeld et al., 2005; Arteaga et al.,*

*2016 and the references therein), however an increase in non–phytoplankton biomass was also necessary to explain the ratios > 1000 observed in this study.”*

Fig. 3e may have also caused confusion because it previously incorrectly displayed maximum POC:Chl-a < 250 even though ratios exceed 1000 (as in Fig. S4d-f). This has been fixed (see our response R2 Line 347-348; line 354-355 below)! We provide additional detail below.

Model outputs from Arteaga et al. (2016, fig. 9) and Behrenfeld et al. (2005, fig. 2, 3) predict maximum phytoplankton POC:Chl-a of ~200 and ~300, respectively, and only in very low productivity regions and seasons, with nearly no available nutrients. Additionally, observation-based studies have found maximum values < 350 (Jakobsen and Markager, 2016, and the references therein), and diatom culture experiments (Laws and Bannister, 1980, from Behrenfeld et al. 2005 fig 3c) reported maximum POC:Chl-a < 500 in their highest nutrient stress and lowest growth rate conditions. Therefore, while phytoplankton-specific POC:Chl-a may have increased during the late-bloom, an increase non-phytoplankton biomass was also necessary to explain the observed ratios.

- R2 Lines 265-269: This part can be moved to the methods.
  We included this short paragraph to give a brief review of how OTUs were defined in our study and provide an explanation for why metazoan sequences were removed from the results presented in this section. This information is also provided with greater detail in the methodology. We have kept this paragraph because it details the final number of OTUs after quality checks and provides context for the rest of the results.

- R1 Line 276: How do you compare the low peridinin concentration with the high dinoflagellate relative abundance?
  We were also initially surprised by the high relative abundance of dinoflagellates in the absence of peridinin. However, the primary dinoflagellates present in the DNA data were most closely related to *Karenia mikimotoi*, which do not use peridinin as their primary pigment. Based on additional accessory pigment analysis, we determined that the relative abundance of dinoflagellates was a result of high gene copy numbers (discussion section 4.4). We go into more detail on the pigment analysis in the supporting material (Supporting analysis, section S2.3).

- R1 Line 336-337: This could be moved to the results.
  We have moved the significance values (Kruskal-Wallis p-values) to the results section 3.2.

  *Updated line 286 → "Biomass specific uptake rates for both C and N ($V_{HCO3\text{-}}$, $V_{NO3\text{-}}$) peaked on different days, but both displayed a significant (Kruskal–Wallis, $p = 6.17 \times 10^{-4}$ and $p = 6.02 \times 10^{-3}$, respectively) increase between days 3 and 4, one day prior to the increase in absolute transport rates (**Fig. 2, S5**).”*

- R2 Fig. 4: Just leaving this comment here with knowing this is hard to change. While reading the paragraph from lines 274-281, I realized I am not able to tell the classes in

[Figure]

*Updated Fig. S7c,d* →

- R1 Figure 4g – This figure is great. It shows changes over time, biological replication, and drivers all in one figure.
  Thank you!

- R2 Line 347-348; line 354-355: These two sentences read a bit like they open a line of reasoning but close it again without giving it enough credit. While the first one says that a depletion of nutrients on day 5 led to the bloom's demise, the last sentence says that factors other than nutrient limitation need to be considered for the bloom's termination. This leaves the question open why the nutrient limitation that is mentioned in the first part is not further discussed.

We agree with the reviewers that the role of nutrient depletion in the bloom's demise should be discussed more deeply. We have updated the discussion of transient nutrient limitation based on the kinetics- and stoichiometry-based thresholding and added a discussion of partial nutrient limitation estimated by the NPZ model.

*Updated lines 400-408 → "However, following the combined kinetics– and stoichiometry–based thresholds outlined in Liang et al. (2019), nutrients were only transiently limiting during the latter half of the bloom (Table S7). There were two timepoints when nutrients were limited in more than one carboy: $SiO_4^{4-}$ on the evening of day 5 and $NO_3^-$ on the evening of day 6, however, neither of these nutrients were limited in back–to–back samples. Additional analysis of the NPZ model revealed that nutrients likely became partially limiting following the bloom peak. Modeled nitrate limitation did not match the timing of the threshold–based limitation, but did reach an average maximum of ~60 % at the end of the incubations (Fig. 3b). The combination of incomplete or transient nutrient limitation and consistently high POM during the late–bloom indicate that factors other than nutrient availability likely contributed to the bloom decline; the potential role of grazing is further investigated below."*

As noted in our response to general comments, we have added the nitrate limitation parameter in the NPZ model ($N_{lim}$, eq. S6) to Fig. 3b and updated the caption accordingly.

[Figure]

*Updated Fig. 3 caption → "**Figure 3:** ... Model outputs are plotted with observations for (a) chlorophyll a, (b) nitrate concentration ([$NO_3^-$]), (c) particulate organic carbon (POC), and (d) particulate organic nitrogen (PON) concentrations. (b) Model–derived nitrate limitation (1-$N_{lim}$, eq. S6) is plotted alongside [$NO_3^-$], such that 0 indicates no*

*growth limitation due to nitrate and 1 indicates complete limitation of growth due to nitrate…"*

- R1 Line 352-355: This is a bit confusing. There only needs to be one limiting nutrient to cause a bloom decline. Here, the authors present evidence for a limiting nutrient on days 5 and 6, which is consistent with when the bloom crashes. It seems nutrients are being prematurely dismissed, but I would argue that they should be given greater focus and discussion in this manuscript.

  We agree that the phrasing, especially "consistently limiting" was a bit confusing. We have updated the text to note that nutrients may have been transiently limiting following the Liang et al. (2019) thresholds – i.e. met criteria for limitation in one sample, but not the following timepoint and not in multiple carboys at the same time, with the exception of the two timepoints already listed in the main text (see our response to R2 Line 347-348; line 354-355 above).

- R2 (Fig. 3) Line 536: "(c) particulate organic carbon (POC) and …" instead of "particulate organic (c) carbon (POC)" in description of Fig. 3.

  Thank you! We have updated the wording as written in the updated figure caption in our response to R2 Line 347-348; line 354-355 above.

- R1 Line 363 – Again, POC:Chl can also be a sign of nutrient limitation. Please review the above references.

  We agree that nutrient limitation likely led to increased phytoplankton POC:Chl-a during the late-bloom, but must conclude that an accumulation of non-phytoplankton biomass was also necessary to explain the observed POC:Chl-a > 1000. As noted above, we have added additional clarification for our reasoning (see response to R1 Line 240 for detailed changes).

- R1 Line 365: Could these metazoan sequences result from copepod detritus?

  It is possible that a portion of the metazoan sequences were from detritus or external DNA. However, if metazoan DNA was primarily present as detritus, we would expect the relative abundance of all metazoan OTUs to be highest at the beginning and decrease over time, as the DNA degraded and the POM of living organisms increased. Instead, our 3 arthropod OTUs have distinct relative abundance patterns, peaking at different points in the bloom.

In the figure below, sp1 is a Maxillopoda spp., sp3 is *Acartia tonsa*, and sp134 is a combination of "other" arthropods.

[Figure]

- R1 Lines 470-477: It may also be worth noting that this incubation was a closed system design and thus likely is unable to capture all the diversity patterns that exist in an open system. A closed system prevents both immigration/emigration and nutrient replenishment which could have impacts on diversity metrics.
  We agree that micro- and mesocosm experiment alpha diversity is bounded by the organisms present in the initial inoculation. We have noted the potential implications of using a closed system experiment in discussion sections 4.3 and 4.6.
  *Updated line 466 → "Open ocean processes, for example mixing, which allows for nutrient replenishment and the introduction of new organisms, would likely cause closed system experiments to deviate from the natural environment on longer timescales or larger study regions. However, robust, short timescale events like blooms are less likely to be disrupted by mixing."*
  *Updated line 555 → "The issue of regionality may occur … because taxa appear more cosmopolitan at small scales (Smith, 2007). The latter effect may be heightened in micro- and mesocosm experiments where diversity is bounded on both ends by the inoculum community."*
- R2 Line 433: When H was already lower than other studies, but not as low as expected during a bloom, were the other studies that are referenced here not during a bloom? Whether they measure H during a bloom or not already makes quite a difference, as also mentioned in the discussion.
  The Wang et al. (2024) and Cram et al. (2024) studies did not specifically target blooms,

but were chosen for comparison because they reported on 18S–based analyses of the whole eukaryotic community in the main stem of the bay.

*Updated line 507 → "The average 18S–based H for the whole community (2.5) was lower than previous non–bloom studies in Chesapeake Bay…"*

Cram et al. (2024) sampled throughout Chesapeake Bay during the summer, while Wang et al. (2024) sampled along both spatial and seasonal gradients. Though Wang et al. (2024) included samples which may have been collected during seasonal blooms, they found no statistical difference in H across seasons (Wang et al., fig. 3) and rarely observed H < 3 (Wang et al., fig. 4).

- R1 Line 470: It's difficult to distinguish the unimodal relationship in the global dataset. Could the points be made transparent (in R, use 'alpha'), to help visualize the density of points?

We have clarified in the main text that the canonical unimodal PDR is defined not only by well delineated curves, but also by clusters of points bounded by a unimodal curve (Smith, 2007). As noted in Skácelová & Lepš (2014), "biomass can be important in regulating the upper limit of diversity, whereas at all the biomass values, extremely species poor communities are found." We have also noted that the well delineated unimodal curve in Irigoien et. al. (2004) showed that phytoplankton- or zooplankton-specific diversity was a function of both phytoplankton and zooplankton biomass as separate parameters. Collapsing Irigoien et al.'s 3D model (2004, fig2c shown below) into two dimensions broadens the range of expected diversity at a given biomass.

[Figure]

*Updated line 547-555 → "While a unimodal relationship, either expressed as a unimodal curve or a cloud of points whose upper bound is defined by a unimodal curve (Smith, 2007; Skácelová and Lepš, 2014), may be observed in larger regional studies, it may not apply to local diversity patterns, which can appear monotonic (Rosenzweig, 1992). This can be seen when comparing the combined global and local data as a whole to individual localized experiments (**Fig. 7a,b**). Excluding the much higher 18S–derived diversity of this study, the upper bounds of all microscopy–derived data clearly displayed unimodal patterns and the data fit flatter, though still significant, unimodal curves. Additionally, Irigoien et al. (2004) showed that phytoplankton– or zooplankton–specific diversity was a*

*function of both phytoplankton and zooplankton biomass as separate parameters, making the unimodal PDR curve of combined factors less distinctly defined. Contrastingly, the individual localized micro- and mesocosm studies displayed both positive and negative monotonic PDRs."*

We added quadratic regression curves fitted to all microscopy-derived diversity data to figure 7a,b by modifying the equation outlined in Irigoien et al. (2004), and noted in the caption and the main text which data is included in the curve, and in the caption that it is representative of 3D model simplified to 2 dimensions. The colors and alpha in figure 7 were changed to make visualization clearer and the colors in Fig. S7 (previously Fig. S6) were changed for consistency (see updated Fig. S7 in our response to R2 Fig. 4 above).

*Updated Fig. 7 →*

[Figure]

*Updated Fig. 7 caption → **Figure 7: Productivity–Diversity Relationship (PDR).** Several studies were compared using 3 different PDRs … Quadratic regression curves, representing a 2D simplification of the Irigoien et al. (2004) 3D model, and linear regressions are plotted for select datasets. Shaded regions around linear regressions indicate the 95 % confidence interval and adjusted $r^2$ and model significance are listed in the respective color of a given dataset. The grey curves in (a,b) are fitted to all microscopy–based diversity data and the grey line in (c) is the linear regression for the combined Atlantic and Chesapeake Bay datasets."*

- R2 Line 476-477: This statement needs some references, I think, even though some are mentioned before in the text.

  We have added an in-text citation for Smith 2007 and clarified that this is for aquatic systems.

  *Updated line 568 → "...negative monotonic relationships are the most common amongst general observations of natural aquatic microbial communities (Smith, 2007)"*

- R1 Line 497: While grazing may have contributed significantly to the observed trends, I still think its important not to discount the role that nutrients may have had on bloom

termination. This is briefly stated on lines 509-511, but could be expanded on throughout.
*Agreed; the impact of nutrient limitation has been noted in the conclusion as well.*
*Updated line 603 → "Nutrient depletion during the late–bloom caused growth rates to decrease and contributed to the bloom's demise, but late–bloom POM accumulation, high POC:Chl–a ratios, and modeled grazing rates indicate that grazing was also necessary to explain the observed trends."*

- R1 References: Sal et al 2013 (Figure 7) is missing from the references.
  *Thank you for catching this. The citation has been added to the references!*
* * *